# Plasma membrane H$^+$-ATPase overexpression increases rice yield via simultaneous enhancement of nutrient uptake and photosynthesis

Maoxing Zhang[1,2,13], Yin Wang[2,3,13], Xi Chen[4,13], Feiyun Xu[1,13], Ming Ding[1], Wenxiu Ye [2,5], Yuya Kawai[6], Yosuke Toda [2,7], Yuki Hayashi[6], Takamasa Suzuki [8], Houqing Zeng[9], Liang Xiao[1], Xin Xiao[10], Jin Xu[11], Shiwei Guo[1], Feng Yan[12], Qirong Shen [1], Guohua Xu [1], Toshinori Kinoshita [2,6✉] & Yiyong Zhu [1✉]

Nitrogen (N) and carbon (C) are essential elements for plant growth and crop yield. Thus, improved N and C utilisation contributes to agricultural productivity and reduces the need for fertilisation. In the present study, we find that overexpression of a single rice gene, *Oryza sativa* plasma membrane (PM) H$^+$-ATPase 1 (*OSA1*), facilitates ammonium absorption and assimilation in roots and enhanced light-induced stomatal opening with higher photosynthesis rate in leaves. As a result, *OSA1* overexpression in rice plants causes a 33% increase in grain yield and a 46% increase in N use efficiency overall. As PM H$^+$-ATPase is highly conserved in plants, these findings indicate that the manipulation of PM H$^+$-ATPase could cooperatively improve N and C utilisation, potentially providing a vital tool for food security and sustainable agriculture.

[1] Jiangsu Collaborative Innovation Center for Solid Organic Waste Resource Utilization, College of Resources and Environment Sciences, Nanjing Agricultural University, Nanjing, China. [2] Institute of Transformative Bio-Molecules (WPI-ITbM), Nagoya University, Nagoya, Japan. [3] Institute of Ecology, College of Urban and Environmental Sciences and Key Laboratory for Earth Surface Processes of Ministry of Education, Peking University, Beijing, China. [4] College of Life Sciences, Nanjing Agricultural University, Nanjing, China. [5] School of Agriculture and Biology, Shanghai Jiao Tong University, Shanghai, China. [6] Graduate School of Science, Nagoya University, Nagoya, Japan. [7] Japan Science and Technology Agency (JST), PRESTO, Kawaguchi, Japan. [8] Department of Biological Chemistry, College of Bioscience and Biotechnology, Chubu University, Kasugai, Aichi, Japan. [9] College of Life and Environmental Sciences, Hangzhou Normal University, Hangzhou, China. [10] College of Resources and Environment, Anhui Science and Technology University, Fengyang, China. [11] College of Horticulture, Shanxi Agricultural University, Taigu, China. [12] Institute of Agronomy and Plant Breeding, Justus Liebig University, Giessen, Germany. [13]These authors contributed equally: Maoxing Zhang, Yin Wang, Xi Chen, Feiyun Xu. ✉email: kinoshita@bio.nagoya-u.ac.jp; yiyong1973@njau.edu.cn

Nitrogen (N) and carbon (C) are indispensable elements for plant growth and are required in large quantities for crop production[1]. Crops largely obtain N from the soil as $NH_4^+$ and/or $NO_3^-$ and C from the atmosphere as $CO_2$. Synthetic N fertilisers are also applied in large amounts, with annual rates of >120 million tons worldwide[2]. Crops have a limited ability to utilise N[3]; thus, excess N is continuously lost from agricultural systems, which pollutes the environment[4]. In addition, the productivity of $C_3$ plants such as rice and wheat is limited by inefficient $CO_2$ fixation by RuBisCO during photosynthesis, due to low $CO_2$ concentrations within the mesophyll cells of leaves. Plant biomass and crop production can be improved by the enhancement of intercellular $CO_2$ concentration, which creates an effect similar to $CO_2$ fertilisation[5], but also emits excess $CO_2$ into the atmosphere[6]. Thus, it is critically important to determine how best to enhance N and $CO_2$ uptake by plants to improve crop production and environmental performance.

Plasma membrane (PM) $H^+$-ATPase, a subfamily of P-type ATPases, generates a membrane potential and $H^+$ gradient across the PM, energising multiple ion channels and various $H^+$-coupled transporters for diverse physiological processes[7,8]. In previous studies, we demonstrated that the PM $H^+$-ATPase mediates light-induced stomatal opening[9,10]. Overexpression of the PM $H^+$-ATPase in guard cells significantly enhances stomatal opening, photosynthesis and, subsequently, growth in *Arabidopsis thaliana*, a model plant[11]. It remains unknown if this manipulation would be efficient in crops, such as rice, which is the staple food for three billion people worldwide[12].

Unlike most terrestrial plants, paddy rice grows in flooded soils where ammonium ($NH_4^+$) ions constitute the dominant N source for root uptake[13]. To use $NH_4^+$ as an N source, rice roots require efficient uptake ability and corresponding assimilation capacity for $NH_4^+$. Conversely, high tissue accumulation of unassimilated $NH_4^+$ is usually negatively correlated with plant growth[14,15]. The assimilation of $NH_4^+$ in root cells requires a C skeleton as the substrate for the synthesis of amino acids through the glutamine synthetase (GS)/glutamate synthase (GOGAT) cycle. The assimilation of one molecule of $NH_4^+$ generates two molecules of $H^+$ in the cytoplasm[16]. PM $H^+$-ATPase facilitates the transport of various nutrients, such as nitrate, phosphate and potassium ($K^+$)[17,18], and maintains cytosolic $H^+$ homeostasis by pumping $H^+$ outside the cells[19]. In our previous study, $NH_4^+$ nutrition was found to induce upregulation of PM $H^+$-ATPase activity in rice roots[20]. Recently, we determined that enhanced PM $H^+$-ATPase activity in rice roots ensures rice growth at high $NH_4^+$ concentrations[21]. Therefore, we hypothesised that PM $H^+$-ATPase may be involved in $NH_4^+$ metabolism in rice plants.

In this study, we examined the involvement of PM $H^+$-ATPase in $NH_4^+$ uptake by rice roots and stomatal opening for $CO_2$ uptake and photosynthesis in rice leaves, with the aim of developing a new strategy to improve rice yield and N use efficiency (NUE) via the overexpression of a single gene, *Oryza sativa* PM $H^+$-ATPase 1 (*OSA1*).

## Results

**PM $H^+$-ATPase mediates $NH_4^+$ absorption**. We first investigated the relationship between PM $H^+$-ATPase and $NH_4^+$ uptake by rice roots. We treated rice roots with the fungal toxin fusicoccin (FC), a stimulator for PM $H^+$-ATPase activity[22], and found that the rate of $^{15}NH_4^+$ absorption increased by 17% in darkness and by 11% under illumination, compared to the corresponding controls (mock) (Fig. 1a). These results clearly indicate that PM $H^+$-ATPase is involved in $NH_4^+$ uptake in rice roots. Under illumination, we also observed an additional increase in the $^{15}NH_4^+$ absorption rate (Fig. 1a) and induction of

leaf stomatal opening for transpiration (Fig. 1b). These results suggest that enhanced transpiration in leaves also contributes to $NH_4^+$ uptake by roots. Therefore, we inferred that the overexpression of PM $H^+$-ATPase in rice roots and/or stomatal guard cells would efficiently improve $NH_4^+$ absorption.

**Phenotype of *OSA1* overexpression and mutation rice lines**. To evaluate the effects of PM $H^+$-ATPase on $NH_4^+$ and $CO_2$ uptake in rice, we focused on a typical PM $H^+$-ATPase isoform, *OSA1*, and investigated the phenotypes of *OSA1*-overexpressing lines (*OSA1*-oxs, driven by the *CaMV-35S* promoter[23], *OSA1#1* to *OSA1#3*) (Fig. 2a), and *osa1* knockout mutants (*osa1-1* to *osa1-3*, *TOS17* insertional mutants) (Fig. 3a and Supplementary Fig. 1). Compared to wild-type (WT) plants, *OSA1* expression was 7.4–8.6-fold higher in roots and 3.5–5.3-fold higher in leaves of *OSA1*-oxs (Fig. 2c), without affecting the expression levels of other PM $H^+$-ATPase isoforms (Supplementary Table 1). *OSA1*-ox plants exhibited ~40% higher $H^+$-ATPase protein levels and ~30% higher PM $H^+$-ATPase activity than did WT plants (Fig. 2d, e), whereas these values were reduced in *osa1* mutants (Fig. 3c–e). We confirmed higher $H^+$ extrusion from roots in *OSA1*-oxs (Supplementary Fig. 2) and proper localisation of overexpressed PM $H^+$-ATPase in roots (Supplementary Fig. 3). When grown under hydroponic conditions, 4-week-old *OSA1*-ox lines exhibited enhanced plant growth, with 18–33% greater dry weight compared to the WT (Fig. 2a, b). By contrast, *osa1* mutants exhibited 33–52% lower dry weight compared to WT plants (Fig. 3a, b). These results indicate that OSA1 (PM $H^+$-ATPase) is a key factor regulating growth in rice. We observed no significant phenotype changes related to growth, relative *OSA1* gene expression, PM $H^+$-ATPase protein levels, stomatal opening (stomatal conductance) and photosynthesis rate in the empty vector-transformed rice under hydroponic conditions (Supplementary Fig. 4).

**Overexpression of PM $H^+$-ATPase enhanced $NH_4^+$ uptake**. To understand the effects of PM $H^+$-ATPase overexpression on N uptake, we compared the isotopic $^{15}N$ ($^{15}NH_4^+$) absorption rate between WT and *OSA1*-oxs (or *osa1* mutants), and determined the absorption rate of $^{15}NH_4^+$ within 5 min by roots. $^{15}NH_4^+$ concentrations ranging from 0.5 to 8 mM were used to test $^{15}NH_4^+$ uptake via different $NH_4^+$ transport systems in rice roots. Interestingly, the $^{15}NH_4^+$ absorption rate in all *OSA1*-oxs was significantly higher than that in the WT, under both low (≤1 mM) and high (≥1 mM) $NH_4^+$ concentration conditions involving the high- and low-affinity transport systems, respectively[24] (Fig. 2f). By contrast, all *osa1* mutants exhibited lower $^{15}NH_4^+$ absorption rates under all $NH_4^+$ concentration conditions (Fig. 3f). We also examined the $^{15}NH_4^+$ absorption rate within 30 min under 2 mM $^{15}NH_4^+$. The rate of $^{15}NH_4^+$ absorption was 20–30% higher in *OSA1*-oxs than in the WT, but was markedly lower in *osa1* mutants (Supplementary Fig. 5). In all rice lines, $^{15}NH_4^+$ absorption rates were significantly repressed by treatment with 0.35 μM vanadate, an inhibitor for PM $H^+$-ATPase (Supplementary Fig. 5). These results confirmed that PM $H^+$-ATPase modification in rice roots regulated $NH_4^+$ absorption. Consequently, under laboratory hydroponic conditions, total N accumulation was found to be 16–57% higher in *OSA1*-oxs (Fig. 2g) but lower in *osa1* mutants (Fig. 3g) compared to the WT. In addition, the contents of other nutrients such as K, P, Ca, S, Fe, and Zn were also increased in *OSA1*-oxs and decreased in *osa1* mutants compared to the WT (Supplementary Fig. 6a–c, f, j). Interestingly, total C accumulation was 21–47% higher in *OSA1*-oxs but lower in *osa1* mutants compared to the WT (Figs. 2h and 3h). Because C is not taken up by plant roots, these results suggest

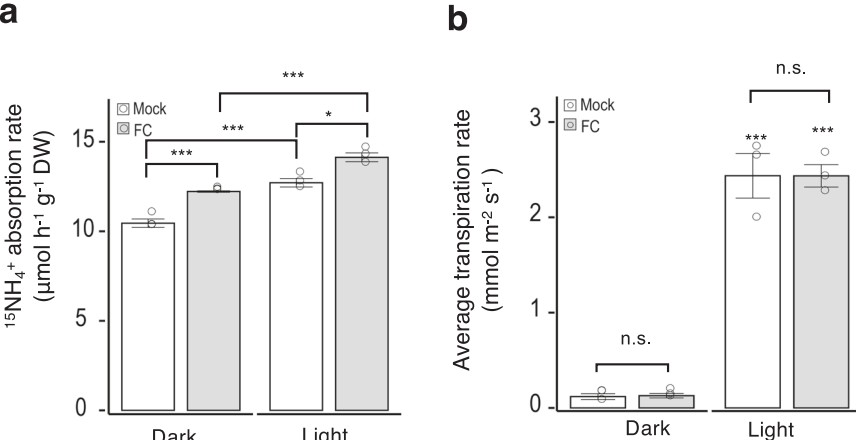

**Fig. 1 Plasma membrane (PM) H$^+$-ATPase regulates ammonium (NH$_4$$^+$) uptake in rice. a** $^{15}$NH$_4$$^+$ absorption rate in wild-type (WT) rice. To determine $^{15}$NH$_4$$^+$ absorption rates, rice seedlings were incubated in 2 mM $^{15}$NH$_4$$^+$ solution with 5 µM fusicoccin (FC) for 30 min under dark or illuminated conditions. **b** Average transpiration rates of rice leaves under dark and illuminated conditions over 30 min. Small circles in **a**, **b** represent data points for individual experiments; three biological replicates were analysed for each treatment. Columns and error bars in **a**, **b** represent the means ± standard errors (SEs; $n = 3$). Differences were evaluated using the two-tailed Student's $t$ test ($^*P < 0.05$; $^{***}P < 0.005$; n.s., not significant). The exact $P$ values are 0.0018 (mock in dark vs. FC in dark), 0.0025 (mock in dark vs. mock in light), 0.0143 (mock in light vs. FC in light) and 0.0016 (FC in dark vs. FC in light) for (**a**); 0.8194 (mock in dark vs. FC in dark), 0.0006 (mock in dark vs. mock in light), 0.0851 (mock in light vs. FC in light), and 4.27 × 10$^{-5}$ (FC in dark vs. FC in light) for (**b**). n.s. Not significant.

that *OSA1* modification influenced CO$_2$ uptake and/or fixation in rice leaves.

**PM H$^+$-ATPase overexpression enhanced stomatal conductance and photosynthetic activity.** Stomata are crucial for gas exchange, particularly for CO$_2$ diffusion into the leaf[12]. Light, the most effective environmental signal for stomatal opening, then activates PM H$^+$-ATPase[10,11,25–28]. PM H$^+$-ATPase-induced hyperpolarisation in the PM of guard cells enables K$^+$ uptake through inward-rectifying K$^+$ channels. The accumulation of K$^+$ and its counter ions in guard cells prompts guard-cell swelling and stomatal opening[29]. Therefore, we investigated stomatal phenotypes in *OSA1*-oxs. Representative closed and open stomata in a WT rice leaf are shown in Fig. 4a. In darkness, the level of stomatal closure in *OSA1*-oxs was similar to that in the WT, whereas under light, the ratio of open to closed stomata was significantly higher in *OSA1*-oxs (Fig. 4b). Conversely, in *osa1* mutants, the ratio of open to closed stomata was significantly lower than in the WT under light treatment (Supplementary Fig. 7a). In all rice lines, stomatal opening was suppressed by the plant hormone abscisic acid (ABA) (Fig. 4b and Supplementary Fig. 7a), suggesting that ABA action was unaffected in guard cells of both *OSA1*-ox and *osa1* mutant plants. Stomatal density, size, and shape in *OSA1*-oxs and *osa1* mutants were comparable to those of the WT (Supplementary Fig. 8), suggesting that overexpression or mutation of PM H$^+$-ATPase in rice had no effect on stomatal morphology or development; these results were similar to our observations in *Arabidopsis thaliana*[12].

Given that stomatal aperture is a limiting factor for photosynthesis[12,30], we examined the photosynthetic properties of *OSA1*-ox plants. Under saturated white light (WL) conditions, stomatal conductance in *OSA1*-oxs was almost double that in the WT (Fig. 4c and Supplementary Table 2), and photosynthetic rates in *OSA1*-oxs were 26–28% higher than in the WT (Fig. 4d and Supplementary Table 2), indicating that enhanced light-induced stomatal opening in *OSA1*-oxs conferred higher photosynthesis rates. By contrast, *osa1* mutants exhibited 22–37% lower stomatal conductance and 27–35% lower photosynthetic rates (Supplementary Fig. 7b, c). Next, we examined photosynthetic light response curves in detail. Along with increased stomatal

conductance (Fig. 4e), the photosynthetic rates of *OSA1*-ox plants were 15–34% higher than those of the WT (Fig. 4f), particularly under high-intensity light (500–1500 µmol m$^{-2}$ s$^{-1}$). Photosynthetic CO$_2$ response curves (*A–Ci* curves) were also higher for *OSA1*-oxs than for the WT (Fig. 4g), indicating a higher photosynthetic capacity among *OSA1*-ox plants. The water use efficiency of *OSA1*-oxs was 13–21% lower than that of the WT (Supplementary Table 2).

**Genome-wide effect of *OSA1* on gene expression.** To identify differentially expressed genes (DEGs) and associated pathways that may provide a molecular basis for the described *OSA1*-ox and *osa1* mutant phenotypes, we analysed the comprehensive gene expression profiles in the leaves and roots of 4-week-old WT, *OSA1*-ox (*OSA1#2*) and *osa1-2* mutant plants using RNA-sequencing (RNA-seq) analysis. Among the DEGs, 1373 and 1124 transcripts were upregulated in the leaves and roots of the *OSA1*-ox line, and 347 and 3295 transcripts were downregulated in the leaves and roots of the *osa1-2* mutant, respectively (Fig. 5a). By contrast, 1895 and 1304 transcripts were downregulated in the leaves and roots of the *OSA1*-ox line, and 1859 and 2913 transcripts were upregulated in the leaves and roots of the *osa1-2* mutant, respectively (Supplementary Fig. 9a–c). Consistent with *OSA1* expression levels, we detected 59 and 82 genes in the leaves and roots, respectively, that were upregulated in the *OSA1*-ox line but downregulated in the *osa1-2* mutant (Fig. 5a).

We then performed Gene Ontology (GO) term enrichment analysis of the DEGs upregulated in the *OSA1*-ox line and downregulated in the *osa1-2* mutant to investigate the molecular mechanisms underlying *OSA1*-mediated biological processes (Supplementary Fig. 9d, e and Supplementary Data 1). The results indicated that 12 biological processes were significantly enriched, including photosynthesis, NH$_4$$^+$ assimilation, glutamate biosynthesis, amino acid metabolism, carbohydrate transmembrane transport, various ion transport, and N utilisation (Supplementary Fig. 9d, e and Supplementary Data 1). Genes associated with transmembrane transporter activity, ion transport, substrate-specific transmembrane transporter activity, cation transmembrane transporter activity, carbohydrate transmembrane transporter activity, and PM part were also

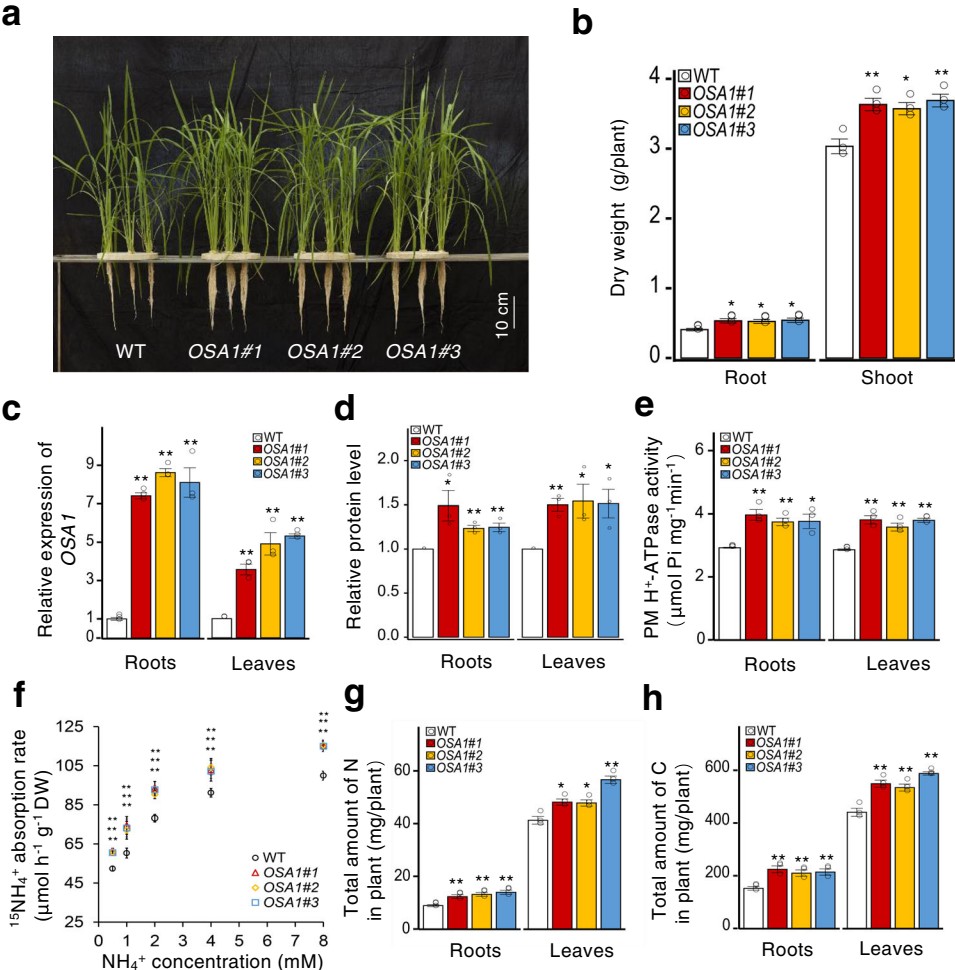

**Fig. 2 *OSA1* overexpression promotes nitrogen (N) and carbon (C) uptake in rice. a** Phenotypes of 4-week-old WT and *OSA1*-overexpressing (ox) plants. **b** Dry weights of WT and *OSA1*-ox plants. **c** Relative *OSA1* expression levels in WT and *OSA1*-ox plants. **d** Relative PM H$^+$-ATPase protein levels in WT and *OSA1*-ox plants. **e** Hydrolytic activity of PM H$^+$-ATPase in WT and *OSA1*-ox plants. **f** $^{15}$NH$_4^+$ absorption rates in the roots of WT and *OSA1*-ox plants under different NH$_4^+$ concentrations. To determine $^{15}$NH$_4^+$ absorption rates, seedlings were incubated with 0.5–8 mM $^{15}$NH$_4^+$ for 5 min to reflect the net uptake rate. **g**, **h** Total N and C levels in WT and *OSA1*-ox plants. Plants were grown hydroponically in a greenhouse for 4 weeks. Small circles in **b–h** represent data points for individual experiments; three biological replicates were analysed for each treatment. Values in **b–h** are presented as the means ± SEs ($n = 3$). Differences were evaluated using the two-tailed Student's *t* test (*$P < 0.05$; **$P < 0.01$). The exact *P* values are provided in the Source Data file.

significantly enriched in overlapping genes that were upregulated in *OSA1*-ox roots and leaves, but downregulated in the *osa1-2* mutant (Supplementary Data 2). In addition, we compared leaf and root transcriptomes between the WT, *OSA1*-ox line, and *osa1-2* mutant, and found that genes associated with nucleic acid binding transcription factor activity, response to chitin, response to organonitrogen compound, regulation of N compound metabolic process, regulation of nucleobase-containing compound metabolic process, and RNA biosynthetic process were significantly enriched in the overlapping genes downregulated in *OSA1*-ox leaves and upregulated in *osa1* mutant leaves (false discovery rate [FDR] < 0.05) (Supplementary Data 3). However, no GO terms were found to be significantly enriched in overlapping genes downregulated in *OSA1*-ox roots and upregulated in *osa1-2* mutant roots (Supplementary Data 3).

A set of genes were enriched in "membrane transport" category. Six NH$_4^+$ transporter genes were enriched in the membrane transport category: *AMT3;3, AMT3;1, AMT2;3, AMT2;1, AMT1;2* and *AMT1.1* (Fig. 5b). These genes encode both high- and low-affinity NH$_4^+$ transporters and were significantly upregulated in the *OSA1*-ox line and downregulated

in the *osa1* mutant in both leaves and roots (Fig. 5b). Transporter genes encoding other cation transporters (e.g. *HAK1, CAX1a* and *CAT1*), electroneutral substance transporters (e.g. *PIP1;3*) or anion transporters (e.g. *PT8*) were also affected by the modification of *OSA1* (Fig. 5b). These results suggest a potential role for *OSA1* in modulating ion and solute transport in plants.

Genes involved in NH$_4^+$ assimilation such as GS (*GS2* and *GS1;2*) and glutamate synthase (*NADH-GOGAT2, Fd-GOGAT2* and *NADH-GOGAT1*) were also strongly affected by the *OSA1* modification (Fig. 5b). Genes associated with photosynthesis were induced by *OSA1* overexpression and repressed by *OSA1* knockout in leaves; these included *Psb28, PsbQ, PsaH, PFPA2, PsbR1, GLO4* and *RbcS* (Fig. 5b).

We examined the expression levels of NH$_4^+$-responsive genes, including *AMT1;1, GS1;2, NADH-GOGAT1, NADH-GOGAT2* and *GS2* using quantitative reverse-transcription polymerase chain reaction (PCR) (Fig. 5d–h). The expression of all investigated genes increased significantly in the *OSA1*-ox lines. Notably, *GRF4*, a key transcription factor in N metabolism and C fixation in rice[31], was highly expressed in response to *OSA1* overexpression (Fig. 5c).

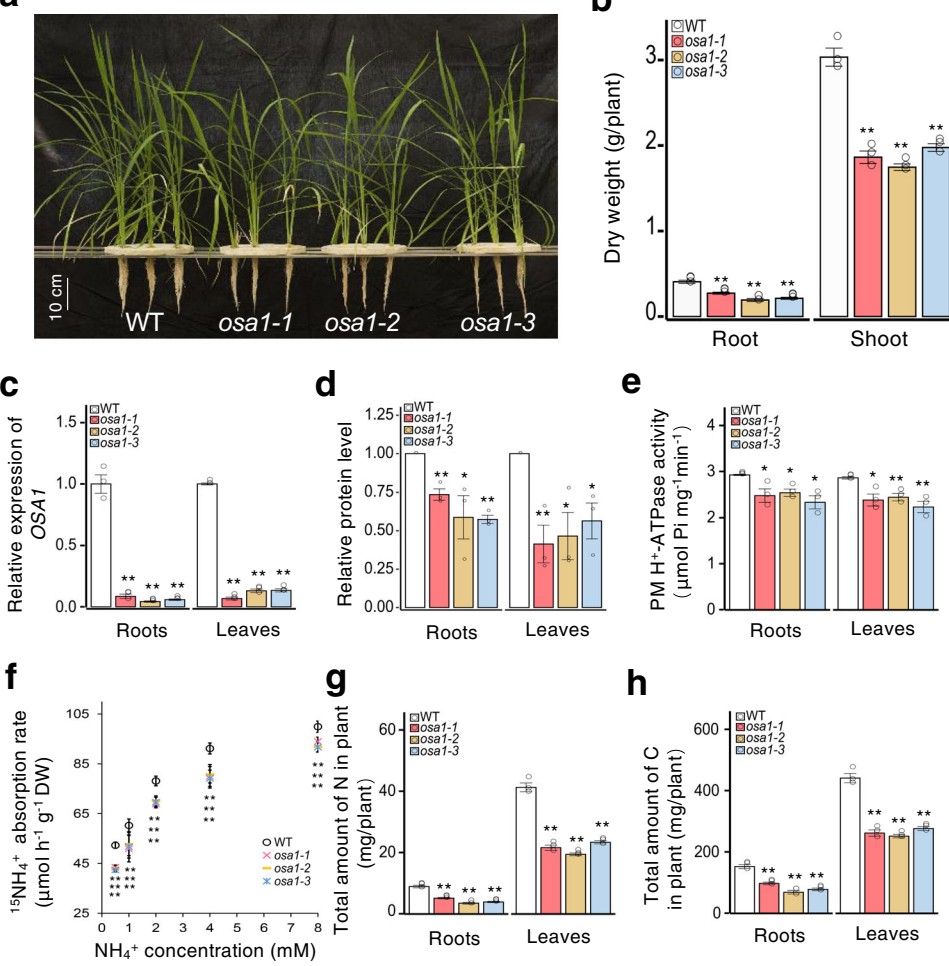

**Fig. 3 osa1 mutant plants exhibit lower N uptake and decreased C content. a** Phenotypes of WT and osa1 plants (scale = 10 cm). **b** Root and shoot dry weights of WT plants and osa1 mutants. **c** Relative expression levels of OSA1 in the roots and leaves of WT plants and osa1 mutants. **d** Relative PM H$^+$-ATPase protein levels in the roots and leaves of WT plants and osa1 mutants. **e** ATP hydrolytic activity of PM H$^+$-ATPase in WT plants and osa1 mutants. **f** $^{15}NH_4^+$ absorption rates in the roots of WT plants and osa1 mutants under different $NH_4^+$ concentrations. The $^{15}NH_4^+$ absorption rate was determined after seedlings were incubated with 0.5–8 mM $^{15}NH_4^+$ for 5 min. **g**, **h** Total N and C levels in roots and leaves of WT plants and osa1 mutants. Plants were grown hydroponically in a greenhouse for 4 weeks; small circles in **b–h** represent data points for individual experiments; three biological replicates were analysed for each treatment. Values in **b–h** are presented as the means ± SEs ($n = 3$). Differences were evaluated using the two-tailed Student's t test (*$P < 0.05$; **$P < 0.01$). The exact P values are provided in the Source Data file.

**Overexpression of PM H$^+$-ATPase promoted field production.** To verify the effects of OSA1 overexpression on rice yield under field conditions, we conducted trials over two growing seasons at three different locations in the middle of China (Nanjing-S in 2016, and Nanjing-N and Fengyang in 2017). We applied urea as an N fertiliser at four different levels: 0 kg ha$^{-1}$ (no N [N–N]), 100 kg ha$^{-1}$ (low N [L–N]), 200 kg ha$^{-1}$ (moderate or normal N [M–N]) and 300 kg ha$^{-1}$ (high N [H–N]). Rice seedlings were planted at a spacing of 25 cm between rows and 20 cm between hills, with a total area of 26 m$^2$ per transect. Stomatal conductance and photosynthetic rates during the vegetative stage exhibited similar trends in the field and laboratory (Supplementary Fig. 10). Representative plants at the reproductive stage under M–N conditions at Nanjing-N are shown in Fig. 6a, b. At all three locations, grain yield of the OSA1-ox lines was 27–39% (mean, 33%) higher than that of the WT (Fig. 6e and Supplementary Tables 3–5). Conversely, in osa1 mutants, grain yield was significantly lower than that of the WT at all three locations (Supplementary Tables 3–5). In OSA1-oxs, the higher yield was correlated with higher panicle weight (18–42%) (Fig. 6f), which

was attributed to increased numbers of panicles per hill (15–20%) (Fig. 6c, g) and spikelets per panicle (8–16%) (Fig. 6d, h). Plant height, panicle length, filled grain rate and 1000-grain weight were nearly identical between OSA1-ox, osa1 mutant and WT plants (Supplementary Tables 3–5). Similar patterns were observed across fertilisation levels (Fig. 6j, Supplementary Fig. 11 and Supplementary Tables 3–5). Notably, under N–N conditions, grain yield was 12–20% higher in OSA1-oxs than in the WT at all test locations (Fig. 6j and Supplementary Tables 3–5). The NUE of OSA1-oxs was ~46% higher than that of the WT at all N fertilisation levels (Fig. 6i). Even when treated with only half the amount of N fertiliser (L–N, 100 kg ha$^{-1}$), the grain yield of the OSA1-ox lines was significantly higher than that of the WT grown under M–N conditions (200 kg ha$^{-1}$) (Fig. 6j). Thus, the same grain yield was attained using only half the amount of N fertiliser when the WT was replaced with OSA1-oxs.

To further verify the practical outcome of OSA1 overexpression in rice, we conducted an independent field trial in Hainan, southern China, which has a tropical climate and short-day conditions, and therefore produces lower yield than the

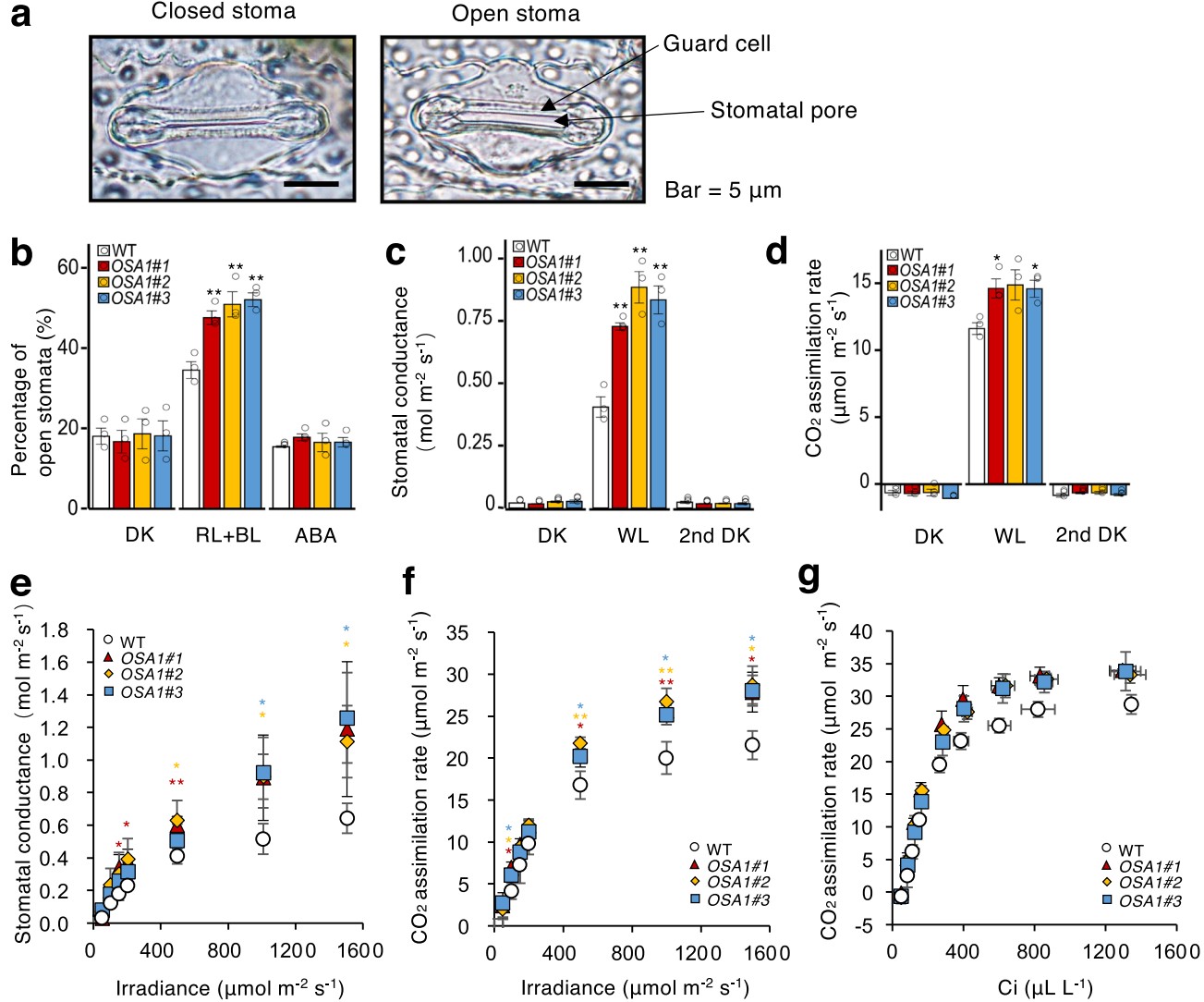

**Fig. 4 Stomatal and photosynthetic properties of *OSA1*-ox plants. a** Representative stomata in the epidermis of WT plants. Experiments were repeated three occasions with similar results. **b** Percentage of open stomata observed after 3 h of darkness (DK), red light plus blue light (RL + BL) or RL + BL in the presence of 20 μM abscisic acid (ABA) in WT and *OSA1*-ox plants (for the details see the "Methods" section). **c, d** Stomatal conductance (**c**) and $CO_2$ assimilation rate under (**d**) DK (30 min), white light (WL; 2 h) and a second DK treatment (30 min) in WT and *OSA1*-ox plants. **e, f** Stomatal conductance (**e**) and $CO_2$ assimilation rate (**f**) in response to light in WT and *OSA1*-ox plants. **g** Relationship between $CO_2$ assimilation rate and intercellular $CO_2$ concentration in WT and *OSA1*-ox plants. Small circles in **b–d** represent data points for individual experiments; three biological replicates were analysed for each treatment. Values in **b–d** are presented as the means ± SEs ($n = 3$) and those in **e–g** are the means ± SDs ($n = 3$). Differences were evaluated using the two-tailed Student's $t$ test (*$P < 0.05$; **$P < 0.01$). The exact $P$ values are provided in the Source Data file.

subtropical areas of central China. Under these conditions, the *OSA1*-ox lines also produced significantly higher grain yield than the WT (Supplementary Table 6).

## Discussion

Increasing crop yield by improving NUE and C fixation is important for sustainable agriculture and environment performance. In this study, we demonstrated the critical role of the PM $H^+$-ATPase gene *OSA1* in controlling both NUE and photosynthesis in paddy rice production. Overexpression of *OSA1* in rice plants increased the activity of PM $H^+$-ATPase (Fig. 2e), promoted $NH_4^+$ uptake and assimilation in roots (Fig. 2f, g) and enhanced light-induced stomatal opening and stomatal conductance and photosynthetic rate under saturated WL in leaves (Fig. 4b–d and Supplementary Table 2), leading to higher NUE and grain yield (Fig. 6). Our results demonstrate the cooperative

enhancement of $NH_4^+$ metabolism, photosynthesis rate and grain yield through the expression modulation of a single PM $H^+$-ATPase gene in rice plants.

PM $H^+$-ATPase was found to regulate $NH_4^+$ uptake in rice (Fig. 1 and Supplementary Fig. 5). Furthermore, genetic evidence based on *OSA1* overexpression/knockout showed that *OSA1* modulation regulated the rate of $NH_4^+$ absorption by rice roots across a wide range of rhizosphere $NH_4^+$ concentrations (Figs. 2f and 3f). RNA-seq analyses revealed the upregulation of at least six $NH_4^+$ transporter genes (*AMT3;3, AMT3;1, AMT2;3, AMT2;1, AMT1;2* and *AMT1.1*) that encode both high- and low-affinity $NH_4^+$ transporters in *OSA1*-overexpressing lines (*OSA1*-oxs), and the downregulation of these genes in the *osa1* mutant (Fig. 5). These results indicate that there is a close relationship between PM $H^+$-ATPase and $NH_4^+$ transporters in rice root cells. These coordinated expression pattern of different genes is also the fundamental mechanisms that enable *OSA1*-oxs rice roots to

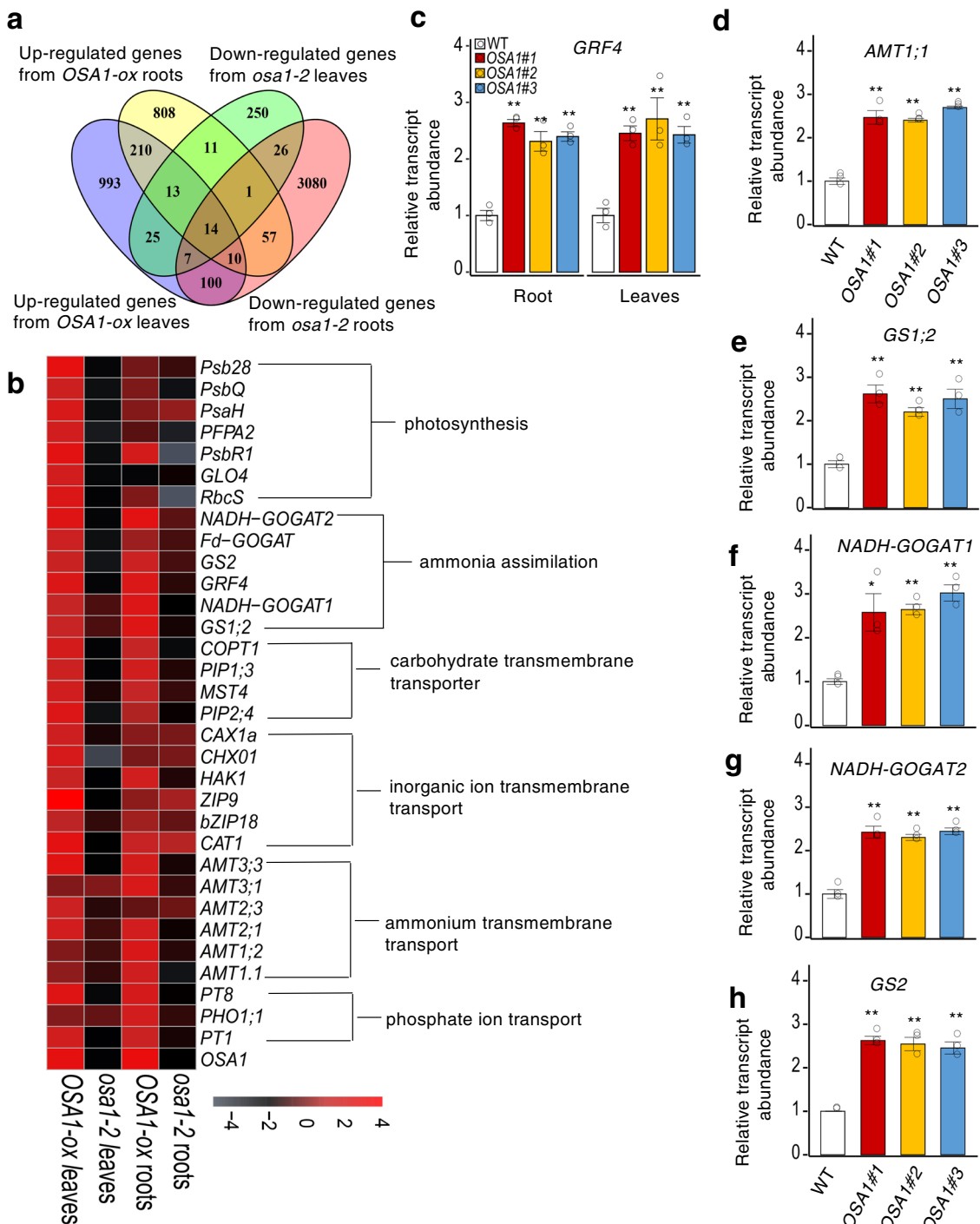

**Fig. 5 Differentially expressed genes (DEGs) enrichment caused by the modification of *OSA1* in rice. a** Venn diagram representing overlapping upregulated genes in *OSA1*-ox and downregulated genes in the *osa1-2* mutant (*osa1*) in roots and leaves (false discovery rate [FDR] < 0.05). **b** Heat map of the DEGs. Significant genes in response to N, C metabolism and ion transport are listed (FDR < 0.05). **c–h** Relative expression of N metabolism-related genes in WT and *OSA1*-ox roots (**c–f**) and leaves (**c, g, h**). Plants were grown hydroponically in a greenhouse for 4 weeks. Small circles in **c–h** represent data points for individual experiments; three biological replicates were analysed for each treatment. Columns and error bars in **c–h** represent the means ± SEs (*n* = 3). Differences were evaluated using the two-tailed Student's *t* test (*P < 0.05; **P < 0.01). The exact *P* values are provided in the Source Data file.

efficiently take up $NH_4^+$ in the field soils with frequently fluctuated $NH_4^+$ concentration. Our results also indicate that *OSA1* overexpression may enhance $NH_4^+$ assimilation capacity. Genes responsible for $NH_4^+$ assimilation such as glutamine synthetase (*GS1;2* and *GS2*) and glutamate synthase (*NADH-GOGAT1*, *NADH-GOGAT2* and *Fd-GOGAT*) were upregulated in *OSA1*-oxs (Fig. 5b, e–h). Because $NH_4^+$ uptake and assimilation are closely

synchronised in plant roots[32], enhanced GS and GOGAT activity can transfer root-absorbed $NH_4^+$ to amino acids for the synthesis of various N-containing compounds during plant growth and development, which in turn prevent $NH_4^+$ overloading in the root cytoplasm due to the acceleration of $NH_4^+$ uptake in *OSA1*-oxs (Fig. 2f). However, the process of $NH_4^+$ assimilation also generates $H^+$, which is toxic if excessively

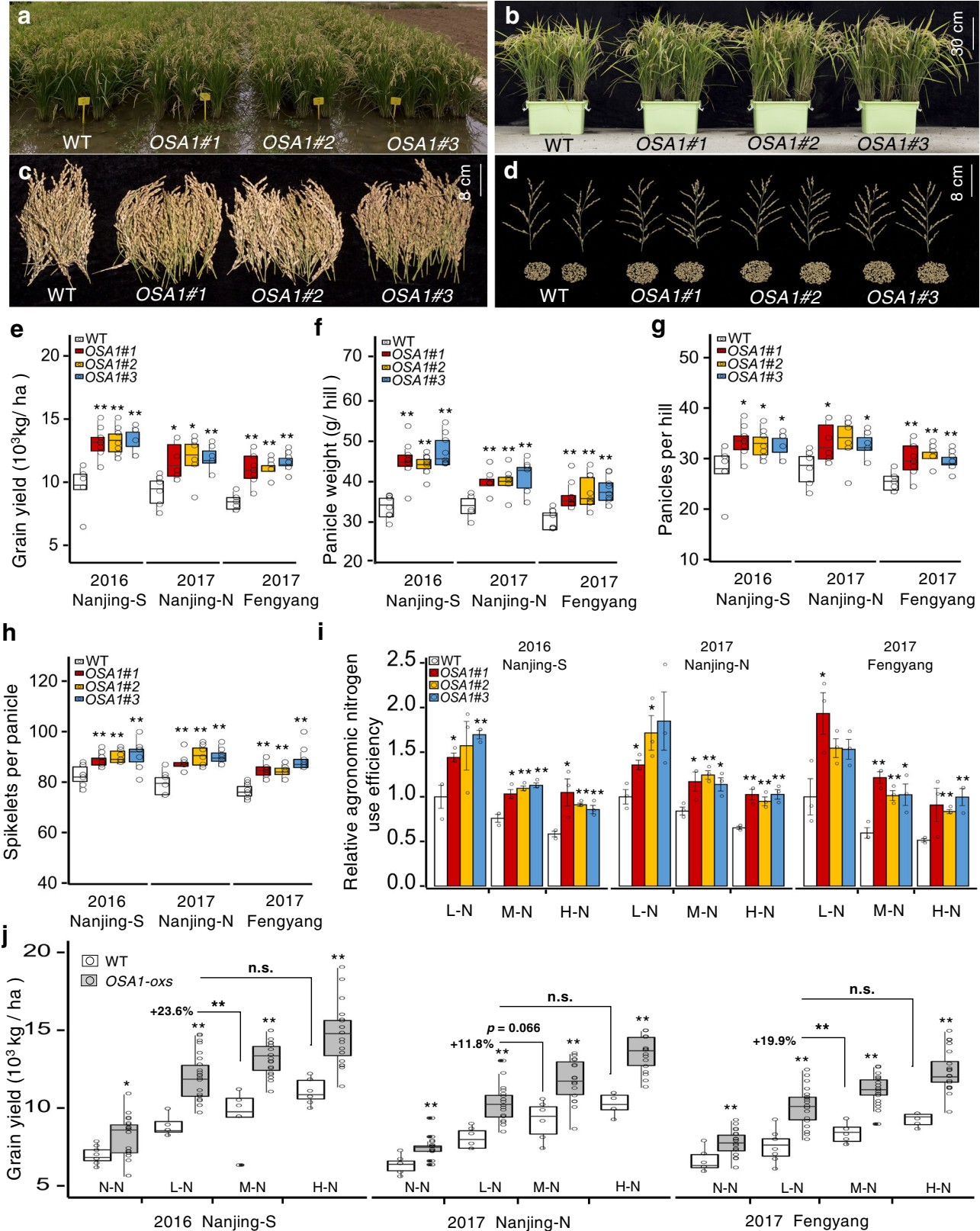

accumulated in the cytoplasm[33,34]. Notably, the overexpression of only $NH_4^+$ transporter genes (*AMT1;1* and *AMT1;3*)[35,36] or glutamine synthetase (*GS1;1* and *GS1;2*)[37] alone led to higher $NH_4^+$ uptake or assimilation rates, but caused poor growth and yields in paddy rice. Considering the important role of PM $H^+$-ATPase in maintaining intracellular pH, enhanced PM

$H^+$-ATPase activity through *OSA1* overexpression (Fig. 2c–e) pumped excessive $H^+$ out of root cells during $NH_4^+$ assimilation (Supplementary Fig. 12). Through this feedback, *OSA1*-ox rice absorbs and uses $NH_4^+$ more efficiently than do WT plants (Figs. 2f–g and 3f–g), which is important for NUE improvement.

**Fig. 6 Overexpression of *OSA1* increases grain yield and N use efficiency (NUE) in the field. a–d** Photographs of 100-day-old WT and *OSA1*-ox plants in the field (**a**), in pots in the field (**b**) and harvested panicles (**c**) and spikelets (**d**) in 2017 in northern Nanjing under 200 kg N ha$^{-1}$ (M–N) fertilisation. **e** Grain yield, **f** panicle weight per plant, **g** panicles per hill and **h** spikelets per panicle of WT and *OSA1*-ox plants in field tests at three locations ($n \geq 6$). **i** Relative agronomic NUE in WT and *OSA1*-ox plants in field tests under low (L–N; 100 kg N ha$^{-1}$), moderate (M–N; 200 kg N ha$^{-1}$) or high (H–N; 300 kg N ha$^{-1}$) levels of N fertilisation. Columns and error bars represent the means ± SEs ($n = 3$). **j** Grain yield of WT and *OSA1*-ox plants in field tests under different N conditions. Black asterisks represent significant differences between WT and *OSA1*-ox plants under the same N fertilisation level; small circles in **e–h**, **j** represent data points of collected samples in individual experiments ($n = 6$ in 2017 Nanjing-N and $n = 8$ in 2016 Nanjing-S and 2017 Fengyang). Centre line indicates the median, upper and lower bounds represent the 75th and the 25th percentile, respectively. Whiskers indicate the minimum and the maximum in the box plots (**e–h**, **j**). Differences were evaluated using the two-tailed Student's *t* test (*$P < 0.05$; **$P < 0.01$; n.s., not significant). The exact *P* values are provided in the Source Data file.

We also observed that *OSA1* is involved in C fixation through the regulation of stomatal opening (Supplementary Fig. 12). Stomatal conductance and photosynthetic rates were enhanced in *OSA1*-oxs due to increased stomatal aperture opening (Fig. 3), compared with rates in the WT and *osa1* mutants (Supplementary Fig. 7). This result is consistent with the finding that genes related to photosynthesis were upregulated by *OSA1* overexpression and downregulated by *OSA1* knockout (Fig. 5b), for example, *Psb28, PsbQ, PsaH, PFPA2, PsbR1, GLO4* and *RbcS*[38–40]. Enhanced photosynthesis in *OSA1*-oxs might also have contributed to the uptake and assimilation of $NH_4^+$ in rice roots by providing more C skeletons and energy for $NH_4^+$ metabolism processes[41]. Therefore, N acquisition and photosynthetic activity are intrinsically linked through overall N and C status in rice plants[42], resulting in a globally coordinated increase in C and N accumulation in rice plants through *OSA1* overexpression (Supplementary Fig. 13). Together, these results show that *OSA1* overexpression promotes both C and N uptake and assimilation, which further regulate the expression of genes involved in C and N metabolism and contribute to plant growth and grain yield.

PM H$^+$-ATPase is also involved in the uptake of various nutrients from plant roots by providing proton motive force[17,18,43]. In this study, the stronger acidification in *OSA1*-ox rice roots (Supplementary Fig. 2) could provide higher proton motive force in the rhizosphere for the uptake of nutrients. This is consistent with the enhanced $NH_4^+$ uptake rate in *OSA1*-oxs as compared with WT plants (Fig. 2f–g and Supplementary Fig. 6), and also consistent with upregulating the expression of various nutrient transporter genes, such as ammonium transporters *AMT1;1/1;2/2;1/3;1/3;3*, phosphate transporter *PT1/PT8/PHO1.1* and potassium transporter *HAK1* in roots of *OSA1*-oxs (Fig. 5b). These results coincided with the increased contents of N, P and K in *OSA1*-oxs as compared with WT plants (Fig. 2g and Supplementary Fig. 6). Recently, GRF4, a transcription factor in rice, was reported to integrate N assimilation, C fixation and plant growth; multiple N metabolism genes, such as *AMT1;1, GS1;2, GS2* and *NADH-GOGAT2*, are positively regulated by GRF4[31]. Here, *GRF4* was found to be upregulated by *OSA1* overexpression (Fig. 5b, c). It is possible that some master regulators of nutrient uptake and metabolism, such as GRF4, could be activated by *OSA1* overexpression. The enhanced C fixation and N metabolism could also have a feedback on the expression of nutrient transporter genes in order to ensure sufficient supply of nutrients for the promotion of the plant growth. Further study is deserved to investigate the underlying molecular mechanisms responsible for the signal transduction initiated by *OSA1* overexpression.

In contrast to CO$_2$, which is taken up from the atmosphere, N is derived from fertilisers for most non-legume crops. Thus, cultivars with improved NUE are in urgent demand for the sustainable development of agriculture. The green revolution has boosted crop yields; however, the resulting cereal varieties are associated with reduced NUE[44]. Even precision crop management has led to only a slight improvement in NUE[3]. In this study,

*OSA1*-ox rice exhibited both higher yield and higher NUE than the WT under a wide range of N fertilisation rates, from 0 to 300 kg N ha$^{-1}$ (Fig. 6i, j and Supplementary Tables 3–5). Higher NUE in *OSA1*-ox rice leads to a lower demand for N fertilisers to produce similar yields of rice grain. To obtain the same yield as WT rice, *OSA1*-ox rice requires only half the amount of N fertiliser (Fig. 6j). This benefit will drastically reduce the cost of rice production as well as the environmental load produced by excess N accumulation due to rice production.

Given that the molecular mechanisms of nutrient uptake[17,43] and light-induced stomatal opening[27] are conserved in most plant species, this manipulation strategy could be applicable to many valuable crops. Therefore, we suggest designating plants overexpressing PM H$^+$-ATPase as **p**romotion and **u**pregulation of plasma **m**embrane **p**roton-ATPase (PUMP) plants. If PM H$^+$-ATPase overexpression can be realised using non-transgenic methods such as genome editing, these crops could have great potential for commercial use, conferring greater yields and potentially critical environmental benefits.

## Methods

**Plant cultivation.** Seeds of WT rice (*Oryza sativa* L. ssp. *japonica* cv. Nipponbare), overexpression lines *OSA1#1, OSA1#2* and *OSA1#3* and mutant lines *osa1-1* (*TOS17* line ND3017), *osa1-2* (*TOS17* line ND3025), *osa1-3* (*TOS17* line ND3033) and CaMV-35S empty vector were surface sterilised in 10% (v:v) H$_2$O$_2$ for 30 min and preincubated in aerated 0.5 mM CaSO$_4$ solution. After 2 days, all seeds were germinated on plastic support nets (mesh size, 2 mm$^2$) floating on 1 mM CaSO$_4$ solution for ~1 week, followed by application of IRRI (International Rice Research Institute) nutrient solution (2 mM NH$_4$Cl, 0.5 mM K$_2$SO$_4$, 0.3 mM KH$_2$PO$_4$, 1 mM CaCl$_2$, 1 mM MgSO$_4$, 0.5 mM Na$_2$SiO$_3$·9H$_2$O, 9 μM MnCl$_2$, 0.39 μM Na$_2$MoO$_4$, 20 μM H$_3$BO$_3$, 0.77 μM ZnSO$_4$, 0.32 μM CuSO$_4$, 20 μM EDTA-Fe) at pH 5.5. For the gas-exchange experiment, seedlings of the WT, *OSA1*-overexpressing lines and *osa1* mutants were grown in ½ IRRI nutrient solution for 1 week, followed by 5 more weeks of growth in modified IRRI nutrient solution (pH 5.5) containing 2 mM NH$_4$Cl. The solution in the containers was replaced every 3 days. Plants for most of the laboratory experiments were grown in a greenhouse at 30 °C/24 °C (day/night) and 60–80% relative humidity. Plants for stomatal aperture and gas-exchange measurements (Fig. 4 and Supplementary Figs. 4 and 7) were grown in a growth chamber (NC-410HC, Nippon Medical & Chemical Instruments Co., Ltd, Osaka, Japan) under ~150 μmol m$^{-2}$ s$^{-1}$ fluorescent light at 30 °C/24 °C (12 h/12 h) and 60–80% relative humidity. The rice seeds for these experiments were of the same age.

For field experiments, plants were grown in the summer of 2016 and 2017 at four well-controlled biological experimental stations in Hainan in 2016 (N18°67′, E108°76′), southern Nanjing in 2016 (N32°01′, E118°51′), northern Nanjing in 2017 (N32°11′, E118°46′) and Fengyang in 2017 (N32°52′, E117°33′). Hainan is in a tropical monsoon zone with sandy soil, whereas the experimental sites in Nanjing and Fengyang are in a subtropical monsoon climate zone with yellow-brown soil. For each field experiment (Fig. 6, Supplementary Fig. 11 and Supplementary Tables 3–6), four levels of N (urea) fertiliser were applied: 0, 100, 200 and 300 kg N ha$^{-1}$ (N–N, L–N, M–N and H–N). Seeds were germinated and seedlings were grown in a greenhouse for ~1 month at the beginning of May. The rice seedlings were hand-transplanted in a flooded field with regular hill spacing. Each fertilisation treatment was performed in one plot (6.5 m × 4 m). Rice seedlings were planted in 14 rows with 20 hills per row, for a spacing of 25 and 20 cm, respectively. Each *OSA1*-ox, *osa1* mutant and WT line was planted in three rows (excluding border hills). At the edge of each plot, the same rice line of inside neighbour was also planted as the border hills (red box) to avoid the margin effects on the rice growth inside the plot. Each plot contained 480 hills, for a total of 1920 hills. Each field experiment consisted of four plots with different N fertilisation levels. Prior to

seedling transplantation, the paddy field was fertilised with 80 kg P ha$^{-1}$ as Ca (H$_2$PO$_4$)$_2$ and 110 kg K ha$^{-1}$ as K$_2$SO$_4$. The first N fertilisation was carried out at 2 days before transplantation using 33.3% of the total amount of N fertiliser, which was mixed into the soil. At the tillering stage (~1 week after transplanting), the second N fertilisation (33.3% of the total N) was carried out. The final N application (33.3% of the total N) was conducted 4 weeks later. The plant growth period (transplantation to harvest) differed among rice lines and N levels, as follows. At 0 or 100 kg N ha$^{-1}$, the growth period was 109 ± 3 days for the WT and *osa1* mutant and 102 ± 2 days for the overexpression lines; at 200 or 300 kg N ha$^{-1}$, the growth period was 119 ± 2 days for the WT and *osa1* mutant and 112 ± 2 days for the overexpression lines.

Grain yield was determined at harvest in October. At maturity, 6–8 hills of plants from each rice line were randomly selected at the centre of the plot from among a 22 × 18 array of hills (excluding the border hills) and harvested. Yield and its components were determined[45,46] with minor modifications. The samples were divided into grain and straw for nutrient content analysis. Agronomic NUE was defined as the yield increase per kg N fertiliser in the field experiment. Relative agronomic NUE (Fig. 6i) was calculated as the ratio to WT rice under L–N treatment in each field trial. Statistical analyses were performed using two-tailed Student's *t* tests and one-way analysis of variance followed by Tukey's test.

**Construction of the overexpression vector and transgenic plants.** The open-reading frame of *OSA1* was amplified using gene-specific primers (Supplementary Table 7). The fragment was treated with restriction enzymes, inserted into vectors and sequenced before transformation. Embryonic rice (*O. sativa*) calli were transformed via *Agrobacterium*-mediated transformation[47]. Three independent homozygous T2 or T3 lines (*OSA1#1*, *OSA1#2* and *OSA1#3*) were used for all phenotypic analyses.

**Quantitative reverse-transcription PCR.** Total RNA was isolated from the roots of WT and transgenic plants using TRIzol reagent according to the manufacturer's instructions[48] (Invitrogen Life Technologies, Carlsbad, CA, USA). Quantitative PCR was performed using an SYBR Premix Ex Taq II (Perfect Real Time) Kit (TaKaRa Biotechnology, Dalian, China) on a Step One Plus Real-Time PCR System (Applied Biosystems, Bio-Rad, CA, USA), and the data were analysed using the $2^{-\Delta\Delta CT}$ method. The *OsActin* and *OsGAPDH* genes were used as internal references to normalise the test gene expression data. All analyses were repeated at least three times. PCR primer sets for gene amplification are listed in Supplementary Table 7.

**Immunodetection.** Leaf and root samples were harvested separately. The samples were immediately homogenised in liquid N and then in ice-cold homogenisation buffer with a mortar and pestle[49]. The membrane proteins were collected by centrifugation and subjected to sodium dodecyl sulfate–polyacrylamide gel electrophoresis and immunoblot analysis; PM H$^+$-ATPase was detected using anti-H$^+$-ATPase antibody[50]. Actin was used as an internal control protein and was detected using anti-actin antibodies (1:3000 dilution, Sigma-Aldrich, St. Louis, MO, USA, Cat#057M4548). Relative PM H$^+$-ATPase levels (Figs. 2d and 3d and Supplementary Fig. 4c) were estimated from the ratio of the signal intensity of PM H$^+$-ATPase to that of actin from the same sample. WT and *OSA1*-ox seedlings were grown in IRRI nutrient solution containing 2 mM NH$_4$$^+$ for 3 weeks in a growth chamber. Immunohistochemical detection of PM H$^+$-ATPase was performed[9] (Supplementary Fig. 3). Roots of 3-week-old seedlings were harvested separately and placed in fixation buffer (4% paraformaldehyde, 60 mM sucrose and 50 mM cacodylic acid; pH 7.4) for 2 h at room temperature. Fixed samples were washed five times with phosphate-buffered saline (PBS) and embedded in 5% agar dissolved in PBS. Sections of 100-mm thickness were prepared using a vibratome (ZERO1; Dosaka EM, Kyoto, Japan) and placed on a glass slide. Samples were treated with enzyme solution (0.1% pectolyase and 0.3% Triton X-100 in PBS) for 2 h, washed five times with PBS, washed once with blocking solution (5% bovine serum albumin) for 10 min and incubated with primary antibody (anti-H$^+$-ATPase) diluted 1000-fold in PBS overnight. On the second day, samples were washed five times with PBS, washed in blocking solution for 10 min and incubated with secondary antibody (Alexa 546, diluted 1000-fold in PBS) for 2 h. Finally, the samples were observed under confocal laser scanning microscopy (FV-10i; Olympus, Tokyo, Japan).

**Measurement of PM H$^+$-ATPase activity.** We determined ATP hydrolytic activity of PM H$^+$-ATPase[51]. Root and leaf tissues were ground in ice-cold homogenisation buffer to isolate the PM in a two-phase partitioning method[51]. PM H$^+$-ATPase hydrolysis activity was determined as the difference between assay results with and without the addition of 0.1 mM Na$_3$VO$_4$ to the reaction solution (Figs. 2e and 3e). The assay was performed in 0.5 mL of reaction solution containing 30 mM BTP/MES, 5 mM MgSO$_4$, 50 mM KCl, 50 mM KNO$_3$, 1 mM Na$_2$MoO$_4$, 1 mM NaN$_3$, 0.02% (w/v) Brij 58, and 3 mM disodium-ATP (substrate for PM H$^+$ ATPase). The reaction was initiated by adding 30 μL of a membrane vesicle suspension containing 1–2 μg total protein and proceeded for 30 min at 30 °C; thus, inorganic phosphate was liberated after the hydrolysis of ATP. The reaction was stopped by adding 1 mL reagent (2% [v/v] concentrated H$_2$SO$_4$, 5% [w/v] sodium dodecyl sulfate and 0.7% [w/v] (NH$_4$)$_2$MoO$_4$), followed by 50 μL 10% (w/v) ascorbic acid. After 10 min, 1.45 mL arsenite-citrate reagent (2% [w/v]

sodium citrate, 2% [w/v] sodium arsenite and 2% [w/v] glacial acetic acid) was added[52]. Colour development was completed after 30 min and measured spectrophotometrically at 720 nm. In each test, H$^+$-ATPase activity was calculated as the amount of phosphate liberated within 30 min mg$^{-1}$ membrane protein in excess of the boiled-membrane protein control.

**$^{15}$N absorption rates in roots of WT, *OSA1*-overexpressing and *osa1* plants.** WT, *OSA1*-ox and *osa1* mutant seedlings were grown in IRRI nutrient solution containing 2 mM NH$_4$$^+$ for 4 weeks in a growth chamber (NC-410HC, Nippon Medical & Chemical Instruments Co., Ltd.) under ~150 μmol m$^{-2}$ s$^{-1}$ fluorescent light at 30 °C/24 °C (12 h/12 h) and 60–80% relative humidity. To determine $^{15}$NH$_4$$^+$ absorption rates within 30 min, seedlings were rinsed in 0.1 mM CaSO$_4$ for 1 min, transferred to modified IRRI nutrient solution containing 2 mM ($^{15}$NH$_4$)$_2$SO$_4$ (atom% $^{15}$N: 98%) incubated with mock 5 μM FC (Fig. 1a) or 350 μM vanadate (Supplementary Fig. 5) for 30 min[53] and rinsed again with 0.1 mM CaSO$_4$ for 1 min[48]. To determine $^{15}$NH$_4$$^+$ absorption rates within 5 min in roots of WT, *OSA1*-ox and *osa1* mutant plants under different NH$_4$$^+$ concentrations, seedlings were incubated with 0.5, 1, 2, 4 and 8 mM $^{15}$NH$_4$$^+$ for 5 min (Figs. 2f and 3f).

Roots and shoots were separated for weighing, and then immediately frozen in liquid N$_2$. After grinding, an aliquot of the powder was dried to a constant weight at 70 °C, and 10 mg of each sample was analysed using the MAT253-Flash EA1112-MS system (Thermo Fisher Scientific, Inc., USA). Each experiment was performed with three independent biological replicates, and statistical analyses were performed using two-tailed Student's *t* tests.

**Nutrient element analysis of plant samples.** Roots and leaves were harvested from 6-week-old plants, washed three times with tap water and rinsed twice (5 min each) with deionised water to remove any adhering nutrients. The leaves and roots were dried in a forced-air oven at 70 °C for ~48 h to a constant weight for dry weight measurements (Figs. 2b and 3b and Supplementary Fig. 4a). The dried samples were ground and passed through a 1.0-mm screen. Total N/C contents (Figs. 2g, h and 3g, h) were determined via the dry combustion method using an Element Analyser (vario EL, Elementar, Langenselbold, Germany). For the analysis of mineral elements, the dry biomass was digested in H$_2$SO$_4$ or HClO$_4$. P concentrations were determined using the molybdate yellow method and K concentrations were determined by flame emission photometry (Supplementary Fig. 6)[54,55]. The other nutrient elements were measured by ICP (Agilent 710 ICP-OES). At least three plants per treatment were harvested, and three independent biological replicates were analysed for each treatment.

**Stomatal observation.** Stomatal observation and quantification were performed[10]. Briefly, epidermal fragments were obtained by homogenising 1-week-old rice seedlings that had been grown in ½ Murashige and Skoog agar medium using a Waring blender. After passing the tissue through a 58-μm nylon mesh, the material was incubated in observation buffer containing 50 mM KCl, 0.1 mM CaCl$_2$ and 5 mM MES-BTP with pH 6.5. After ~3-h incubation in darkness or light (150 μmol photon m$^{-2}$ s$^{-1}$ red light [LED-R; EYELA] plus 50 μmol photon m$^{-2}$ s$^{-1}$ blue light [Stick-B-32]) in 20 μM ABA, epidermal fragments were collected for microscopic observation. The percentage of open stomata (Fig. 4b and Supplementary Figs. 7a and 8c) was quantified as the number of open stomata per total stomata observed. At least 100 stomata were observed per treatment; three biological replicates were analysed for each treatment, and statistical analysis was conducted using two-tailed Student's *t* tests.

**Gas-exchange measurements.** Gas-exchange measurements were performed using the LI-6400 System (Li-Cor) with a standard chamber. Light and CO$_2$ response curves were constructed based on data obtained using measurement processes and light sources[12,56]. The flow rate, leaf temperature and relative humidity were kept constant at 500 μmol s$^{-1}$, 24 °C and 60–75% (Pa/Pa), respectively. Under each light/CO$_2$ condition, photosynthetic rate and stomatal conductance data were collected after these values reached a steady state (15–30 min). Fully expanded leaves from 6-week-old plants were used in these experiments (Figs. 1b and 4 and Supplementary Figs. 7b, c and 4d, e). WL was provided by a fibre optic illuminator with a halogen projector lamp (15 V/150 W, Moritex, San Jose, CA, USA) as a light source powered by an MHAB-150W (Moritex) power supply. For CO$_2$ response curves, leaves were measured at saturating WL conditions (~1500 μmol m$^{-2}$ s$^{-1}$) (Fig. 4g and Supplementary Fig. 7b, c). To obtain the stomatal conductance and CO$_2$ assimilation rate data shown in Fig. 4c, d and Supplementary Table 2, leaves were measured under saturating WL conditions (~1000 μmol m$^{-2}$ s$^{-1}$).

For field gas-exchange measurements (Supplementary Fig. 10), the flow rate of the Li-6400 system was kept constant at 500 μmol s$^{-1}$ at a leaf temperature and relative humidity of 28 °C and 40–50% (Pa/Pa), respectively. All measurements were performed before the heading stage. At least three plants were selected for measurement, and three biological replicates were analysed for each treatment.

**Stomatal density and size.** Three to four fully expanded leaves of 6-week-old rice plants were selected. At least five microphotographs were randomly taken of the

adaxial or abaxial surface of the leaf lamina. The average stomatal density and size (long axis of each stoma) were calculated[57] (Supplementary Fig. 8a, b).

**High-throughput RNA-seq analysis.** For RNA-seq analysis, total RNA was extracted from leaves and roots collected from 4-week-old WT, *OSA1*-ox (*OSA1#2*) and *osa1* (*osa1-2*) mutant rice lines using a TRIzol Plus RNA Purification Kit (Thermo Fisher Scientific, Waltham, MA, USA). Complementary DNA libraries were constructed using a TruSeq RNA Sample Prep Kit v. 2 (Illumina, San Diego, CA, USA) and sequenced using a NextSeq 500 system (Illumina). Base-calling of sequence reads was performed using the NextSeq 500 pipeline software. Only high-quality sequence reads (50 continuous nucleotides with quality values >25) were used for mapping (Fig. 5a, b and Supplementary Fig. 9). Reads were mapped to *O. sativa* (IRGSP v. 1.0 2019.8.29) transcripts using the Bowtie software[58]. Experiments were repeated three times separately. We obtained 10.1–13.6 million sequence reads per experiment. Gene expression values were reported in RPM (reads per million mapped reads) units. Normalisation of read counts and statistical analyses were performed using the EdgeR software package[59,60] and the Degust Ver. 3.1.0 web tool (http://degust.erc.monash.edu). Obtained RPM values were further analysed using MS Excel software. Only genes with $\log_2$ fold change $\geq 1$ or $\leq -1$, and an FDR < 0.05 were considered to be significant DEGs. GO term enrichment was conducted using GO Term Enrichment tool in the Plant Transcriptional Regulatory Map (Plan-tRegMap) website[61] (http://plantregmap.gao-lab.org/go.php). GO category (http://geneontology.org/) FDR ≤ 0.05 was regarded as significantly enriched.

**Detection of rhizosphere acidification in roots.** Rhizosphere acidification in WT and *OSA1*-oxs roots was determined[51]. The roots of 7-day-old plants were thoroughly washed with deionised water and spread on an agar sheet containing 0.7% (w/v) agar, 0.02% (w/v) bromocresol purple, 2 mM $NH_4Cl$ and 1 mM $CaSO_4$ at pH 5.6. The roots were carefully pressed into the agar to avoid damage. For visualisation of rhizosphere acidification, incubation was conducted in a growth chamber in the dark for 12 h. The relative area of rhizosphere acidification (Supplementary Fig. 2b) was estimated as a ratio to the WT area (yellow area on agar sheet; Supplementary Fig. 2a). At least three plants per treatment were harvested, and three independent biological replicates were analysed for each treatment.

**Quantification of $H^+$ extrusion rate.** The $H^+$ efflux rates from the WT and *OSA1*-ox rice roots were measured using the scanning ion-selective electrode technique (SIET System BIO-003A, Younger USA Science and Technology Corp., Applicable Electronics Inc., Science Wares Inc., Falmouth, MA, USA) (Supplementary Fig. 2f)[15,21]. Briefly, seedlings were placed in 50 mL of growth solution with 2 mM $NH_4^+$ for 12 h. Then, the rice roots of 7-day-old plants were washed with deionised water and equilibrated in the measuring solution for 10 min. The equilibrated roots were then transferred to a measuring chamber, which contained 3 mL of a solution comprising 0.2 mM $CaCl_2$, 0.1 mM KCl, 0.1 mM $NaNO_3$ and 0.5 g $L^{-1}$ MES (2-morpholinoethanesulfonic acid sodium salt) (pH 5.7). At least three plants per treatment were analysed, and three independent biological replicates were performed.

**Reporting summary.** Further information on research design is available in the Nature Research Reporting Summary linked to this article.

## Data availability
The authors declare that the data supporting the findings of this study are available within the paper and the Supplementary information. The RNA-seq data that support the findings of this study have been deposited in the DNA Data Bank of Japan (DDBJ) with the accession number DRA011260. Source data are provided with this paper.

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

## Acknowledgements

This work was supported by grants from the National Key Basic Research and Development Program (2017YFD0200200/0200206 to Y.Z.), the Natural Science Foundation of China (NSFC 31471937 to Y.Z.), Technology of Japan and the Advanced Low Carbon Technology Research and Development Program from the Japan Science and Technology Agency (JPMJAL1011 to T.K.) and the Natural Science Foundation of Anhui Province, China (1608085MC59 to X.X.), as well as Grants-in-Aid for Scientific Research on Innovative Areas (15H05956, 20H05687 and 20H05910 to T.K.).

## Author contributions

Y.W., T.K. and Y.Z. conceived the research project and designed the experiments. M.Z., Y.W., F.X., M.D., Y.K., Y.T., Y.H., T.S., H.Z., L.X., X.X., S.G., T.K. and Y.Z. conducted experiments and performed data analyses. Y.W., X.C., W.Y., J.X., F.Y., Q.S., G.X., T.K. and Y.Z. oversaw the entire study and wrote the manuscript.

## Competing interests

The authors declare no competing interests.
