## [Peer Review File · Nature Communications]

Reviewers' comments:

Reviewer #1 (Remarks to the Author):

The manuscript of Zhang et al describes the potential relationship between PM (H+) ATPase activity, ammonium transport and carbon balance in rice. Data presented suggests activation/deactivation of ATPase activity (phosphorylation or through genetic means) influences ammonium uptake, we also presume ammonium assimilation with final outcomes delivering larger plants and increased seed yields. Lab-based and field based experiments have been conducted with WT and transgenic lines where increasing N provision has been provided to explore relationships with potential gains in nitrogen-use efficiencies. Importantly, this work follows on from previous studies which have shown that the PM (H+) ATPase (OSA1) influences light-induced guard cell opening rice and arabidopsis which enhance PS activity and potentially the ability to deliver increased C flow to the plant. This is exciting research highlighting new possibilities to enhance plant productivity and nitrogen use efficiencies.

A conclusion proposed in this manuscript is that OSA1 activity is linked to ammonium transport. Increased or decreased OSA1 activity influences the plants ability to grow and develop seed yields, respectively. Although there is some evidence in the manuscript to suggest ammonium uptake increases in both artificially phosphorylated PM (H+) ATPase (WT plants) or in over expression lines (OSA1-3), the direct proof that increased ammonium uptake is responsible for this collective response still remains elusive. There is no other analysis presented to suggest that other nutrients (K+, Pi, metals etc) or water for that matter are not influencing the growth and yield responses being highlighted. We may be observing a general response of the plants to a more efficient nutrient uptake pathway and assimilatory process with increased nitrogen driving the size of the response. I suggest the following controls: 1) test for growth on an alternative nitrogen source (nitrate) to challenge the ammonium specific claim; 2) test for changes in other ions (roots and shoots) which may also be increasing in response to enhanced ATPase activities and 3) complete a more detailed ammonium and or nitrate flux analysis using the OSA and osa mutants to show enhanced/decreased uptake across the LATS or HATS concentration ranges, this should be matched with 15N pulse-chase flux analysis to show ammonium transport specifically is enhanced in roots or in other tissues where ammonium flux is required and where OSA activity is observed.

A second conclusion rests on the plants ability to deliver increased carbon flow through greater stomatal aperture and PS rates. This data is interesting and highlights that enhanced C availabilities may improve growth when reduced N is available. However, it is unclear how exactly this is linked to ammonium and PM (H+) ATPase activities. Is more C flowing to seeds and does the C flow mirror the N dose responses used in the field studies (see below). As discussed above, it would be nice to show this relationship is with ammonium nutrition specifically (e.g. ammonium increases tiller numbers and yield while nitrate or enhanced Pi doesn't).

I have a number of questions regarding the data presented and the experimental designs.

- 1) Fig 1. There are no mock controls under light
- 2) Fig 1. The ammonium flux is 30 min, which will involve significant efflux out of the root as well as redistribution to the shoot. The 15N absorption rates may be less than they should be?
- 3) All Figures. The statistical analysis needs to be clarified. In figure 1 is this data from three independent experiments (multiple reps each) or just three plants as indicated by the circles? Does Fusicoccin influence other OSA activities?
- 4) The flux rates are considerably higher from Figures 2 onward. Is there a reason for this due to the experimental conditions?
- 5) Fig 4 and 5. There is little description and or follow up on the white light (WL) treatments in the result and discussion sections?
- 6) The field trials need to be described in more detail. A description of the design, plot size, statistical design and climatic conditions. When the data presented in Fig 6 (6e-i) is compared to

that of 6J, it would suggest quite different responses in the field and a general N response occurring only in the latter experiment (6j). The calculations of agronomic NUE are not well defined. Data in 6i would suggest NUE should be going down instead of up as there is no real yield bounce as external N levels increase from 0-300 kg/Ha.

7) The field trials and lab-based experiments should include a test cultivar which is an adapted current variety used for the region. I would expect that if this is a dominant trait, prior breeding should have already captured this trait. Furthermore, what natural variation exists for this trait - does it already exist in adapted lines?

8) The description of the second field experiment in Southern China is limited.

9) What is the status of each OEX of KO line. Are these all isogenic to the WT being compared? Furthermore, the use of a WT plant is fine, but to be fair to the transgenic process, an empty vector transformed control should be included in the analysis or at least demonstrated they are the same as the WT plants.

Reviewer #2 (Remarks to the Author):

The authors describe the overexpression of a H⁺-ATPase in rice that results in improved seed yield through improve N and C utilization. This is an important finding with putative agronomic applications. However, there are still some issues the authors should take into account.

-the title is too general for the presented data: the data only show one example where the simultaneous N and C uptake increases yield. To my opinion, it is better to mention the gene and not over-interpret the data.

- the abstract is not entirely clear: how should one interpret an average increase of 33%? The increase in seed yield is 39%, so there must be aspects of yield that are less (maybe even negatively?) affected? In addition, there is a big leap between the presence of homologs in other species and the fact that this strategy would also work in other species in the field.

- There is too few information on the methodology to judge how solid the results are. Some examples: the immunodetection is described, but it is not mentioned which antibodies were used. Without the specifics of the antibodies, this experiment is hard to assess. The 'same rice lines' were planted as border: is this all three transgenic lines? Is it the wild type? I don't fully understand how the calculations were done for the liberated phosphate: was the same tissue used? Were the calculations normalized for biomass? Often ratios are determined: how were the statistics done for the ratios? For the field trial it is clear how many plants were planted, but not on how many plants the observations were done: on all?

- my biggest issue with the manuscript is the controls for the phenotypic and molecular analyses. All comparisons were done relative to a wild type nipponbare. Since the rice lines were generated through agrobacterium-mediated transformation of calli a tissue culture step is involved, which typically can give rise to somaclonal variations. To avoid this, it would have been nicer to work with segregating wild types. In this way, the authors would also have been able to avoid effects due to the age of the seedstock, the different conditions for seed upscale (Temperature etc largely affect seed quality). Now, it can not be concluded if a difference in seed quality and age is not responsible for the observed phenotypes.

- I'm not a stomata expert, but I'm puzzled by Fig.4: Is the right panel of the OE line (legend suggests that) or an example of an open stoma (figure suggests that)? More evidence should be provided to conclude that there is no difference in morphology. The text now states that because there is no difference in density and size that there is no effect on morphology: I do not fully agree: morphology can be altered while density stays the same.

- the data are very descriptive: we OE a gene and this is what we see, without providing

mechanistic insights. The authors attempt to do so by providing new data in the discussion, but this is limited (and should be in the results section and not in the discussion).

Reviewer #3 (Remarks to the Author):

The agricultural green revolution of the 1960's enhanced cereal crop yields, fed a growing world population, and was in part due to increased cultivation of semi-dwarf GRVs. However, GRVs require a high N fertilizer supply to achieve maximum yield potential. Because environmentally degrading nitrogen fertilizer use underlies current cereal yields, it is thus imperative to develop new strategies for increasing nitrogen use efficiency (NUE). In this manuscript, the authors reported that the up-regulation of the OSA1 gene could enhance carbon fixation, nitrogen assimilation and biomass production, and consequently increased grain yield in rice. The topic is interest for researchers follow trends and important development in understanding the molecular mechanisms underlying maintaining the balance of carbon (C) and nitrogen (N) in response to environmental changes. However, a number of aspects of the manuscript are not clear and several experiments need to be improved.

1, the previous studies have shown that overexpression of the genes encoding plasma membrane (PM) H⁺-ATPase promoted stomatal opening, photosynthetic C-fixation, and plant growth. In the current version of the manuscript, the authors generated the *osa1* mutant and the transgenic rice plants overexpressing of OSA1, and confirmed that OSA1 acts as a positive regulator of C-fixation, N assimilation and plant growth in rice. However, the authors only present the observations, but did not provide the molecular mechanisms that explain how OSA1 coordinatedly regulates C fixation in shoots, N assimilation in roots and biomass production.

2, to investigate if OSA1-enhanced transpiration in leaves accelerates NH₄⁺ uptake in roots and grain yields, the authors could generate the OSA1-overexpressing plants under the control of guard cell-specific promoters.

3, in Figs. 4b and 5a, the authors showed that OSA1-mediated stomatal opening was completely suppressed by ABA, indicated a link between OSA1 function and ABA signalling. In canonical ABA signalling, PP2Cs interact with and inactivate OST1, ABA binds RCAR/PYR/PYL receptors to capture PP2Cs, releasing the inhibition of OST1, consequently promoting activities of the slow-type anion channel SLAC1 and other targets in guard cells. The authors should analyze the genetic relationship between OSA1 and the key components of ABA signalling, such as OST1 and PP2C.

4, in Fig. 1, the authors should add the un-treated control samples (mock) under light conditions.

5, in Fig. 6, the authors should present in more details about agronomic traits.

6, the authors performed qRT-PCR experiments and showed the up-regulation of those genes involved in N-metabolism, it will be better that the authors carried out the RNA-seq experiments using WT, *osa1* and OSA1-overexpressing plants.

Reviewer #4 (Remarks to the Author):

The manuscript "Simultaneously enhancing N and C uptake drastically increases in rice yield" describes very interesting and important discoveries in Agriculture and human society. Authors Wrote that only one gene, H-ATPase overexpression can improve crop productivity due to N and C uptake improvement.

Major

It is convincing from your manuscript that Membrane hyperpolarization due to H ATPase can improve N and C uptake and as a result improve biomass.

However, I think that increase of seed production (increase of # of panicle per hill and spiklets per panicle) is not directly related with biomass increase and explain it. Even though you can not explain the mechanism. You should describe the changes of flowering and panicle development genes expression through transcriptome analysis.

Reply to Reviewers' comments:

Thank you very much for your valuable comments and suggestions. We have revised the manuscript according to the reviewer's comments and have answered all the critical questions. All changes to the text are written in red letters.

Reviewer #1 (Remarks to the Author):

The manuscript of Zhang et al describes the potential relationship between PM (H^+) ATPase activity, ammonium transport and carbon balance in rice. Data presented suggests activation/deactivation of ATPase activity (phosphorylation or through genetic means) influences ammonium uptake, we also presume ammonium assimilation with final outcomes delivering larger plants and increased seed yields. Lab-based and field based experiments have been conducted with WT and transgenic lines where increasing N provision has been provided to explore relationships with potential gains in nitrogen-use efficiencies. Importantly, this work follows on from previous studies which have shown that the PM (H^+) ATPase (OSA1) influences light-induced guard cell opening rice and arabidopsis which enhance PS activity and potentially the ability to deliver increased C flow to the plant. This is exciting research highlighting new possibilities to enhance plant productivity and nitrogen use efficiencies.

A conclusion proposed in this manuscript is that OSA1 activity is linked to ammonium transport. Increased or decreased OSA1 activity influences the plants ability to grow and develop seed yields, respectively. Although there is some evidence in the manuscript to suggest ammonium uptake increases in both artificially phosphorylated PM (H^+) ATPase (WT plants) or in over expression lines (OSA1-3), the direct proof that increased ammonium uptake is responsible for this collective response still remains elusive.

1. There is no other analysis presented to suggest that other nutrients (K^+ , Pi, metals etc) or water for that matter are not influencing the growth and yield responses being highlighted.

Response: Thank you for your suggestion. We analyzed the contents of other two macronutrients, K and P, in the rice plants, and found they were both significantly higher in OSA1-overexpressing lines (OSA1-oxs) than in wild type, but lower in *osa1* mutants (Supplementary Fig. 8). It is most likely that OSA1 functioned in facilitating the uptake of other nutrients, since PM H^+ -ATPase plays a central role in energizing the transportation of various nutrients (Palmgren et al. 2001). We described these results in the text as follows.

Page 6, line 124-126; In addition, the contents of P and K, two important macro-nutrients, were also increased in OSA1-oxs and decreased in *osa1* mutants (Supplementary Fig. 8a, b) when compared to wild type (WT).

The transpiration rate of OSA1-oxs was shown to be increased due to the higher stomatal opening in the leaves (Supplementary Table 2). The higher transpiration rate in OSA1-oxs may facilitate the acquisition of water with various nutrients by roots, which could also improve plant growth.

2. We may be observing a general response of the plants to a more efficient nutrient uptake pathway and assimilatory process with increased nitrogen driving the size of the response.

Response: In this study, we found the role of *OSA1* in increasing NUE and grain yield by cooperatively enhancing ammonium uptake and assimilation and carbon fixation. Our field trials with different application levels of nitrogen fertilizer showed that the N nutrition was the major effect associated with grain yield of rice in our study (Fig. 6j). Because the application of P and K fertilizers were at the same level with enough amount in all treatments, which were not the limiting for rice growth. The enhanced uptake of other nutrients, such as P and K, by *OSA1* overexpression may amplify nitrogen-driving the size of the response.

3. I suggest the following controls: 1) test for growth on an alternative nitrogen source (nitrate) to challenge the ammonium specific claim;

Response: Thank you for your suggestion. According to your suggestion, we have cultivated *OSA1*-oxs and WT with nitrate as the sole nitrogen source and found that *OSA1*-oxs also have higher biomass than WT (Unpublished Fig. 1A). Because nitrate uptake is dependent on the proton motive force, it is possible that nitrate uptake by plant roots could be increased by the enhanced activity of PM H⁺-ATPase mediated by *OSA1* overexpression. However, as compared with ammonium nutrition, the growth of rice plants under nitrate nutrition in hydroponic cultivation system was significantly repressed (Unpublished Fig. 1B). The leaves of all the rice seedlings including *OSA1*-oxs showed chlorosis phenotype when cultivated under sole nitrate nitrogen condition (Unpublished Fig. 1A). The possible reason is that paddy rice, especially Japonica rice cultivar used in this study, prefers ammonium N source rather than nitrate. The nitrate-induced rice leaf chlorosis has also been demonstrated previously (Chen et al. 2018 *Plant Cell Environ.*). Due to the growth disorder of rice plants under sole nitrate nitrogen condition, thus it may be not suitable to analyze the phenotypes like leaf stomatal opening, photosynthesis rate and field trials by using nitrate as the sole N source.

Unpublished Fig. 1 (A) WT and three *OSA1*-ox lines were cultivated in nutrition solution containing 2 mM NO_3^- as a nitrogen source for 14 days after germination. Nitrogen is NO_3^- (2mM). The pH of nutrient solution is adjusted to 5.5. (B) Comparison of the rice growth under treatment of 2 mM NO_3^- or NH_4^+ .

Chen, et al. 2018. *H₂O₂ mediates nitrate induced iron chlorosis by regulating iron homeostasis in rice.* *Plant Cell Environ.* 41: 767-781

4. 2) test for changes in other ions (roots and shoots) which may also be increasing in response to enhanced ATPase activities

Response: Thank you for your suggestion. As we answered #1, we have analyzed the contents of the other two macronutrients, K and P, and found they were both significantly higher in *OSA1*-oxs than that in WT (Supplementary Fig. 8).

5. and 3) complete a more detailed ammonium and or nitrate flux analysis using the *OSA* and *osa* mutants to show enhanced/decreased uptake across the LATS or HATS concentration ranges, this should be matched with ^{15}N pulse-chase flux analysis to show ammonium transport specifically is enhanced in roots or in other tissues where ammonium flux is required and where *OSA* activity is observed.

Response: Thank you for your important suggestion. According to your comments, we have conducted the experiment again to analyze NH_4^+ absorption rate under a wide range of NH_4^+ concentration (0.5–8 mM) (Kronzucker, H.J. et al. 1998 *Plant Cell Physiol.*). In order to minimize the NH_4^+ efflux, we limited the incubation time of roots to 5 min. As shown in Fig. 2f & Fig. 3f in the revised manuscript, overexpression of *OSA1* increased the influx of NH_4^+ either under the concentration ranges of high affinity transport system (HATS) or low affinity transport system (LAPS) (Fig. 2f), while mutation of *OSA1* depressed the influx of NH_4^+ (Fig. 3f). We described these results in the text as follows.

Page 6, line 6-13; The absorption rate of $^{15}\text{NH}_4^+$ within 5 min by roots was determined. $^{15}\text{NH}_4^+$ concentrations ranged from 0.5 to 8 mM were used for testing $^{15}\text{NH}_4^+$ uptake via different NH_4^+ transport systems in rice roots. Interestingly, the absorption rate of $^{15}\text{NH}_4^+$ by *OSA1*-oxs lines was always significantly higher than that by WT either under high affinity transport system ($\leq 1\text{m M}$) or low affinity transport system ($\geq 1\text{m M}$)²⁴ (Fig. 2f). In

contrast, *osa1* mutants always showed decreased absorption rate of $^{15}\text{NH}_4^+$ under all concentrations (Fig. 3f).

Kronzucker, H.J. et al., 1998. Dynamic interactions between root NH_4^+ influx and long-distance N translocation in rice: insights into reed back processes. Plant Cell Physiol. 39: 1287-1293

6. A second conclusion rests on the plants ability to deliver increased carbon flow through greater stomatal aperture and PS rates. This data is interesting and highlights that enhanced C availabilities may improve growth when reduced N is available. However, it is unclear how exactly this is linked to ammonium and PM (H^+) ATPase activities. Is more C flowing to seeds and does the C flow mirror the N dose responses used in the field studies (see below). As discussed above, it would be nice to show this relationship is with ammonium nutrition specifically (e.g. ammonium increases tiller numbers and yield while nitrate or enhanced Pi doesn't).

Response: This is true in our study that the improved rice growth and grain yield under reduced N fertilization in *OSA1*-oxs are due to the enhanced stomatal opening and PS rate (Fig. 6j and Supplementary Fig. 5). As shown in Supplementary Fig. 4, both the stomatal conductance and CO_2 assimilation rate were higher in *OSA1*-oxs, in comparison with WT, under reduced N fertilization in the fields. It is also clear that the stomatal conductance and CO_2 assimilation is positively related to the N availability in the fields (Supplementary Fig. 4). As a result, enhanced accumulation of C in grains is correlated to the enhanced accumulation of N for *OSA1*-oxs compared to WT and *osa1* mutants (Supplementary Fig. 7c). This result provided evidences that more C flowing to seeds mirror the N dose responses used in the field studies. Taken together, it is clear that it is linked to ammonium and PM H^+ ATPase.

In our field trials (Supplementary Tables 3-6), the tiller number and grain yield of WT and *OSA1*-oxs were all enhanced by the increased application rate of N fertilizer (urea, which is converted to ammonium in soils). In addition, at the same application rate of N fertilizer, the tiller number and grain yield of *OSA1*-oxs were both significantly higher than those of WT. Furthermore, the carbon content in *OSA1*-oxs seeds was always higher than in WT seeds (Supplementary Fig. 7c), indicating that more carbon had been flowed to the seeds by overexpression of *OSA1* to reach the higher grain yield under different N fertilization levels. Interestingly, the ratios of C/N contents in seeds of WT, *OSA1*-oxs, and *osa1* mutants were similar (Supplemental Fig. 7c), suggesting that C flow should have mirror the N dose response in the field studies.

Nitrate was not used as a N fertilizer in our field trials. Because in agricultural practices, nitrate is less stable in paddy soils, compared to ammonium, and is easily lost through leaching, runoff or denitrification. Due to the inhibition of nitrification by anaerobic condition in paddy soils, the content of nitrate is usually low in paddy soils. Although rice may have taken up nitrate to some extent, due to the possible nitrification in the rhizosphere by excretion of O_2 through aerenchyma in rice roots. According to the great difference between the amount of ammonium by application of urea fertilizer and possible available nitrate in paddy soils, it is most likely that the increases of tiller number and grain yield by N fertilizer are related to ammonium nutrition and may not come from the effect of nitrate.

The amount of P-fertilizer was used at the same levels for all the N-fertilizer treatments in our field trials. Thus, the levels of P should be nearly the same in the paddy soils for all the N-fertilizer treatment. The increase of tiller number and grain yield by N fertilizer should be mainly related to ammonium nutrition, and may not come from the effect of P. Our results are also consistent with the Tanaka and Garcia (1964) that tiller formation of rice depends largely on the nitrogen absorbed and the carbohydrates produced at the growth stage when the tiller primordium grows.

Although we cannot exclude the functions of *OSA1* overexpression in improving the uptake of nitrate and P, but the higher grain yield of *OSA1*-oxs and NUE under low nitrogen and high nitrogen conditions is associated with enhance ammonium uptake and assimilation (Supplemental Fig. 10).

Tanaka, A. & Garcia, C.V. 1964. Studies of the relationship between tillering and nitrogen uptake of the rice plant. Soil Sci. Plant Nutr. 11: 31-37

I have a number of questions regarding the data presented and the experimental designs.
7. Fig 1. There are no mock controls under light

Response: We have repeated this experiment with the Mock control under light. In addition, Fig.1 is now reorganized in the revision, in order to make the relationship between NH_4^+ absorption rate and PM H^+ -ATPase or transpiration rate more clearly than the previous version. In this way, the redundant data were omitted in this revision, since the activation/phosphorylation of PM H^+ -ATPase by fusicoccin has been convinced over 20 years before (Marrè 1979 Ann. Rev. Plant Physiol., Olsson et al. 1998 Plant Physiol.)

Marrè, E. 1979. Fusicoccin: a tool in plant physiology. Ann. Rev. Plant Physiol. 30: 273-288

Olsson, A. et al. 1998. A phosphothreonine residue at the C-terminal end of the plasma membrane H^+ -ATPase is protected by fusicoccin-induced 14–3–3 binding. Plant Physiology. 118.2: 551-555.

8. Fig 1. The ammonium flux is 30 min, which will involve significant efflux out of the root as well as redistribution to the shoot. The ^{15}N absorption rates may be less than they should be?

Response: We agree with you. Our results represented the net absorption rate of $^{15}\text{NH}_4^+$ within 30 min. We determined the total $^{15}\text{NH}_4^+$ amount in the whole plants, the $^{15}\text{NH}_4^+$ translocated to other tissues like leaves was also included. Supposed that the permeability of plasma membrane of root cells to NH_4^+ is same in WT rice, our results revealed that fusicoccin treatment caused the significant increase of the NH_4^+ absorption rate in comparison with the control.

According to your suggestion, we have conducted another experiment to analyze the $^{15}\text{NH}_4^+$ influx by decreasing the incubation time to 5 min (Fig. 2f and Fig. 3f) (Kronzucker, H.J. et al. 1998 Plant Cell Physiol.). As we described in Question 5, the overexpression of *OSA1* increased the influx of NH_4^+ either under the concentration ranges of high affinity transport system (HATS) or low affinity transport system (LAPS) (Fig. 2f), while mutation of *OSA1* depressed the influx of NH_4^+ (Fig. 3f).

Kronzucker, H.J. et al., 1998. Dynamic interactions between root NH_4^+ influx and long-distance N translocation in rice: insights into reed back processes. *Plant Cell Physiol.* 39: 1287-1293

9.1 All Figures. The statistical analysis needs to be clarified. In figure 1 is this data from three independent experiments (multiple reps each) or just three plants as indicated by the circles?

Response: We used three representative plants for each treatment in Fig. 1 and repeated 3 times with independent experiments. In addition, we have repeated the many similar experiments of $^{15}\text{NH}_4^+$ uptake again in this study, such as Fig. 2f, Fig. 3f and Supplementary Fig 6. We have clarified the statistical analysis in the Figure legends and Methods.

9.2 Does Fusicoccin influence other OSA activities?

Response: It has been demonstrated that PM H^+ -ATPase is activated by phosphorylation of a penultimate residue, threonine, and binding of 14-3-3 protein to the phosphorylated PM H^+ -ATPase (Palmgren 2001 *Annu Rev Plant Physiol Mol Biol*, Kinoshita and Shimazaki 1999 *EMBO J*), and that FC irreversibly activates PM H^+ -ATPase via stabilizing binding of 14-3-3 protein to the phosphorylated PM H^+ -ATPase (Marré 1979 *Annu Rev Plant Physiol*, Kinoshita and Shimazaki 2001 *Plant Cell Physiol*). Rice genome has been reported to encode 10 OSAs (Baxter et al. 2003 *Plant Physiol*). Actually, OSA1-OSA9 possess a putative phosphorylated penultimate threonine residue, but OSA10 not. In addition, expression analysis revealed that OSA10 is hardly expressed in whole plants (Toda et al. 2016 *Plant Cell Physiol*). Therefore, it is most likely that FC influences most of OSA gene products expressed in plants.

Palmgren, M.G. 2001. Plant plasma membrane H^+ -ATPase: Powerhouses for nutrient uptake. *Annu. Rev. Plant Physiol. Plant Mol. Biol.* 52: 817-854

Kinoshita, T. & Shimazaki, K. 1999. Blue light activates the plasma membrane H^+ -ATPase by phosphorylation of the C-terminus in stomatal guard cells. *EMBO J.* 18, 5548 – 5558

Marré, E. 1979. Fusicoccin: A tool in plant physiology. *Ann. Rev. Plant Physiol.* 30, 273-288

Kinoshita, T. & Shimazaki, K. 2001. Analysis of the phosphorylation level in guard-cell plasma membrane H^+ -ATPase in response to fusicoccin. *Plant Cell Physiol.* 42, 424-432

Baxter. et al. 2003. Genomic comparison of P-type ATPase ion pumps in Arabidopsis and rice. *Plant Physiol.* 132, 618-628

Toda, Y. et al. 2016. *Oryza sativa* H^+ -ATPase (OSA) is involved in the regulation of dumbbell-shaped guard cells of rice. *Plant Cell Physiol.* 57, 1220 – 1230

10. The flux rates are considerably higher from Figures 2 onward. Is there a reason for this due to the experimental conditions?

Response: This is because the incubation time of roots in this experiment is 5 min, less than the incubation time of previous experiment (30 min) (Fig. 1a). When we calculate the flux rate, the incubation time should be considered. As we known, the influx rate will decrease with the incubation time, while the efflux will increase with the incubation time (Zhu et al., 2005. *Plant Cell Physiol.*). At the beginning, the influx rate is much higher than efflux rate, then the efflux rate will increase with time, and finally the rates of influx and efflux will become constant. So, if the incubation time is shorter, the flux rate would be higher. This is why the absorption rate in Fig. 2 f is higher than the result in Fig. 1a.

Zhu, Y, et al., 2005. A link between citrate and proton release by proteoid roots of white lupin (Lupinus albus L.) grown under phosphorus-deficient conditions? Plant Cell Physiol. 46: 892-901

11. Fig 4 and 5. There is little description and or follow up on the white light (WL) treatments in the result and discussion sections?

Response: Thank you for your suggestion. We added the descriptions regarding the white light treatment, as follows.

In the result, P. 7 line 150 - P. 8 line 156: Under saturated white light (WL) condition, the stomatal conductance in *OSA*-oxs was almost twice higher than WT (Fig. 4c and Supplementary Table 2) and the photosynthetic rate in *OSA1*-oxs lines was around 26-28% higher than WT (Fig. 4d and Supplementary Table 2), indicating that enhanced light-induced stomatal opening in *OSA1*-oxs conferred higher rates of photosynthesis. In contrast, *osa1* mutants showed around 22-37% lower stomatal conductance and around 27-35% lower photosynthetic rate (Supplementary Fig. 2b, c).

In the discussion, P.12 line 239-242; enhanced light-induced stomatal opening and stomatal conductance and photosynthetic rate under saturated WL in leaves (Fig. 4b-d and Supplementary Table 2)

12. The field trials need to be described in more detail.

12.1 A description of the design, plot size, statistical design and climatic conditions.

Response: Thank you very much for your important suggestions. We have added the description of design, plot size, statistical design and climatic conditions for the field trials in Methods of the revised MS (P. 16 line 331 - P. 18 line 365).

12.2 When the data presented in Fig 6 (6e-i) is compared to that of 6J, it would suggest quite different responses in the field and a general N response occurring only in the latter experiment (6j).

Response: The traits related to grain yield in Fig. 6 e-h were collected from the results under treatment of N fertilization at 200 kg N/ha (moderate or normal N conditions, M-N) from three different locations. The results under all treatments of N fertilization at other levels, such as N-N, L-N, H-N, were shown in Supplementary Fig. 5 and Supplementary Tables 3-5. The relative NUE and grain yields under all the treatments of different N fertilization rate from three locations were summarized and shown in Fig.6i and j.

12.3 The calculations of agronomic NUE are not well defined. Data in 6i would suggest NUE should be going down instead of up as there is no real yield bounce as external N levels increase from 0-300 kg/Ha.

Response: Thank you for your suggestion. We added more details on the calculations of agronomic NUE (Page18 Line 361 - 365). We have normalized NUE of WT under low N condition (L-N) to 1 and calculated other NUEs of *OSA1*-oxs under different N level. Now, it is obvious that the NUE is reduced with the increase of N application rate. The NUE of *OSA1*-oxs is higher than that of WT at any application rate of N fertilizer and the results were similar in the field trials of three different locations (Fig. 6i).

13. The field trials and lab-based experiments should include a test cultivar which is an adapted current variety used for the region. I would expect that if this is a dominant trait, prior breeding should have already captured this trait. Furthermore, what natural variation exists for this trait - does it already exist in adapted lines?

Response: In this study, rice cultivar (*Japonica* cv. *Nipponbare*) was used because its genome sequence is released, its genetic transformation is practicable, and the knockout mutants are available. However, it is difficult for us to use an adapted regional variety, because its genetic transformation may not be practicable. We have conducted the field trials in four different locations in China (Nanjing-south, Nanjing-north, Fengyang, and Hainan; Fengyang is in the North of Nanjing, about 190 km far away from Nanjing. Hainan is the Southmost province of China). The effects of *OSA1* overexpression on the grain yield and NUE were similar when the rice plants were cultivated in any of these locations, confirming the role of *OSA1* in improving grain yield NUE in different locations.

We have checked the SNPs of *OSA1* in many rice cultivars (<https://snp-seek.irri.org/>), and there is no sequence variation for many of the adapted rice cultivars in China, like Zhongchao123, Yunguang8, Nangeng46, and Yandao9. Some SNPs positions exist in other cultivar. In the future study, it would be very interesting to investigate whether the rice varieties with high yield and/or NUE have higher activities of PM H⁺-ATPase in their roots and leaves.

14. The description of the second field experiment in Southern China is limited.

Response: Thank you for your suggestion. We have described in the revision, Page11 Line 228 -232 and Page 16 Line 334 – Page 17 Line336.

15.1. What is the status of each OEX of KO line. Are these all isogenic to the WT being compared?

Response: We used T2 or T3 homozygous OSA1-overexpressing lines (Page 18 Line 371-372) and homozygous OSA1-knockout mutants (Supplementary Fig.1). Yes, OSA1-overexpressing lines and knockout mutants are all isogenic to the WT.

15.2 Furthermore, the use of a WT plant is fine, but to be fair to the transgenic process, an empty vector transformed control should be included in the analysis or at least demonstrated they are the same as the WT plants.

Response: Thank you very much for your important suggestion. According to your suggestion, we have prepared the empty vector transformed control line and compared the growth, stomatal opening (stomatal conductance), and photosynthesis rate between the empty vector transformed control and WT plants, as much as we could. We found there is no difference between them (Unpublished Fig. 2).

Unpublished Fig. 2. Plant growth and gas-exchange properties in CaMV 35S empty vector transformed rice. (a) Dry weights of root and shoot in WT and CaMV 35S empty vector transformed plants. Stomatal conductance (b) and CO₂ assimilation rate (c) in response to light in WT and CaMV 35S empty vector transformed plants. Error bars represent the SD (n = 3) (a to c). Differences were assessed using the Student's *t*-test (**P* < 0.05; ** *P* < 0.01).

Reviewer #2 (Remarks to the Author):

The authors describe the overexpression of a H⁺-ATPase in rice that results in improved seed yield through improve N and C utilization. This is an important finding with putative agronomic applications. However, there are still some issues the authors should take into account.

16. the title is too general for the presented data: the data only show one example where the simultaneous N and C uptake increases yield. To my opinion, it is better to mention the gene and not over-interpret the data.

Response: Thank you for your important suggestion. We have modified the title as follows. "Overexpression of plasma membrane H⁺-ATPase *OSA1* increases rice grain yield via simultaneously enhancing N and C uptake".

17.1 the abstract is not entirely clear: how should one interpret an average increase of 33%? The increase in seed yield is 39%, so there must be aspects of yield that are less (maybe even negatively?) affected?

Response: Thank you for pointing this out. The 33% is the mean of grain yield increases (27–39%) of *OSA1*-overexpressing lines from three field trials in three different locations. We have revised it, as follows.

P.2 line 32 - 34 "As a consequence, overexpression of *OSA1* in rice plants caused 33% increase in grain yields and 46% increase in nitrogen use efficiency in fields."

17.2 In addition, there is a big leap between the presence of homologs in other species and the fact that this strategy would also work in other species in the field.

Response: Thank you for your important suggestion. Actually, we tested plant growth in the field using only rice, as shown in the present paper. However, we found that overexpression of PM H⁺-ATPase in Poplar showed 20% faster plant growth than wild type with higher transpiration and photosynthesis rates in the growth room (Unpublished Figure 3), as well as *Arabidopsis* (Wang et al. PNAS 2014) and rice in the present paper. In addition, it has been demonstrated that PM H⁺-ATPase is known to be important for nutrient uptake (Palmgren, 2001 Annu Rev Plant Physiol Mol Biol), and that overexpression of PM H⁺-ATPase in *Medicago* plants increased phosphate uptake in roots (Wang et al., 2014 Plant Cell). Taken together, it is most likely that overexpression of PM H⁺-ATPase is useful for enhancement of plant growth, stomatal conductance, photosynthetic activity, and nutrient uptake in other plant species in addition to *Arabidopsis*, rice and Poplar. We have revised the description in Discussion, as follows.

P. 14 line 304- 306 "Given that the molecular mechanisms for nutrient uptake and light-induced stomatal opening are conserved in the most of plant species, this manipulation strategy could be applicable to many valuable crops."

Wang, Y. et al. 2014. Overexpression of plasma membrane H⁺-ATPase in guard cells promotes light-induced stomatal opening and enhances plant growth. *Proc. Natl. Acad. Sci. USA* 111, 533–538.

Palmgren, M.G. 2001. Plant plasma membrane H⁺-ATPase: Powerhouses for nutrient uptake. *Annu. Rev. Plant Physiol. Plant Mol. Biol.* 52: 817-854

Wang, E. et al. 2014. A H⁺-ATPase that energizes nutrient uptake during mycorrhizal symbioses in rice and *Medicago truncatula*. *Plant Cell* 26, 1818–1830 .

Unpublished Fig. 3 (A) Comparison of plant growth between wild type Poplar (T89) and plasma membrane H⁺-ATPase overexpressing lines (GC1::AHA2). (B) Pictures of Poplar plants 80 days after transfer to the soil.

18. There is too few information on the methodology to judge how solid the results are. Some examples: the immunodetection is described, but it is not mentioned which antibodies were used.

Response: Thank you for your suggestion. Regarding the antibody information for immunodetection, we have added descriptions in the Methods section, as follows.

P19, line 388 - line 392, "PM H⁺-ATPase was detected with anti-H⁺-ATPase antibody as described previously⁵⁴. Actin as the internal control protein was detected with anti-actin antibodies (1:3,000 dilution, Sigma-Aldrich, Cat#057M4548). Relative PM H⁺-ATPase level (Fig. 2d, and Fig. 3d) was estimated from the ratio of the signal intensity of PM H⁺-ATPase to that of actin from the same sample."

19. Without the specifics of the antibodies, this experiment is hard to assess.

Response: The antibody used for detection of OSA1 protein was raised against to the conserved catalytic domain of the PM H⁺-ATPase (Hayashi et al., 2011 *Plant Cell Physiol*). Therefore, it is possible that the antibodies recognize all isoform of PM H⁺-ATPase isoforms. However, to check specific increase of OSA1 in OSA1-overexpressing lines and decrease of OSA1 in knockout lines, we performed qRT-PCR in all OSA1-overexpressing lines and knockout lines (Fig. 2c, Fig. 3c, and Supplementary Table 1). The results showed that expression levels of other isoforms were not affected in OSA1-overexpressing lines and knockout lines (Supplementary Table 1). From these results, we conclude that increase of PM H⁺-ATPase protein level in

OSA1-overexpressing lines (Fig. 2d) and decrease of PM H⁺-ATPase protein level in knockout lines (Fig. 3d) are caused by the change of OSA1 protein amount.

Hayashi, M. et al. 2011. Immunohistochemical detection of blue light-induced phosphorylation of the plasma membrane H⁺-ATPase in stomatal guard cells. *Plant Cell Physiol.* 52, 1238-1248 .

20. The 'same rice lines' were planted as border: is this all three transgenic lines? Is it the wild type?

Response: As shown in Unpublished Fig. 4, we planted the same rice lines as its neighbor inside on the border (Border hills) around the tested rice lines (test hills).

Unpublished Fig. 4 Schematic diagram of rice planting in field experiments. Each fertilization treatment has one individual plot with 4 x 4 m in size. Rice seedlings were planted in 14 rows with 20 hills per row, that means rows and hills were 25 cm and 20 cm apart respectively. The border hills on the red line were planted using same rice lines as their neighbor inside to avoid the margin effect on the rice growth inside the plot. Shade area represents the rice plants, that were used to determine the yield and other parameters (the tested hills).

21. -I don't fully understand how the calculations were done for the liberated phosphate: was the same tissue used? Were the calculations normalized for biomass?

Response: Thank you for your suggestions. In this study, we performed ATP hydrolysis assay using isolated plasma membrane from roots or leaves. PM H⁺-ATPase hydrolyzes ATP into ADP and Pi (inorganic phosphate) and liberated phosphate in 30 min was used to represent the hydrolysis activity of PM H⁺-ATPase. We repeated 3 times using biologically independent materials and calculated the average. The calculations were normalized by mg protein in plasma membrane. We have revised the description in Methods (Page 20, line 409 - 411).

22. -Often ratios are determined: how were the statistics done for the ratios?

Response: Thank you for your important suggestion. In our original manuscript, we have mentioned ratios in several positions. These ratios were used to represent relative gene expression levels, relative PM H⁺-ATPase protein level, percentage of open stomata and relative agronomic N use efficiency (NUE), etc.

We have revised them and added more details on the calculations and statistical analysis of ratios.

The first is the ratio of opened stomata. Here, the ratio is a generalized ratio, which tell us the percentage of opened stomata number based on the total stomata number observed (Fig. 4b and Supplementary Fig. 2a). In order to make reader easy to understand, we have used percentage instead of ratio in results (Page 7, line 138 - 142). Three biological replicates were performed, and statistical analysis was conducted by two-tailed student's *t* test (Page 21, line 452 - Page 22, line 456).

The second ratio represents the relative agronomic N use efficiency (NUE) (Fig. 6i). NUE of all the rice lines under different N fertilization levels are divided by the NUE of WT under L-N in each field tests. Please see Page 18 Line 361- Line 365. Statistical analyses were performed using two-tailed Student's *t* test.

Relative gene expression levels (Fig. 2c, Fig. 3c, and Fig. 5c-5h) were determined by real-time quantitative PCR analysis. The relative amount of PCR product was quantified using the comparative cycle threshold method and normalized to the two internal control genes, *OsActin* and *OsGAPDH* (Page 18, lines 380-381). Statistical analyses were performed using two-tailed Student's *t* test.

Relative H⁺-ATPase level (Fig. 2d, and Fig. 3d) was estimated from the ratio of the signal intensity of PM H⁺-ATPase to that of actin from the same sample in Page 19, line 390-392. Statistical analyses were performed using two-tailed Student's *t* test.

23.-For the field trial it is clear how many plants were planted, but not on how many plants the observations were done: on all?

Response: Thank you for your suggestion. We have added more information on the plant cultivation in the Methods section. Please look at Unpublished Fig. 4. Each fertilization treatment has one individual plot with 4 x 4 m in size. Rice seedlings were planted in 14 rows with 20 hills per row, that means rows and hills were 25 cm and 20 cm apart respectively. Each line of OSA1-oxs and WT were planted in three rows (excluding the border hills). On the edge of the plot, same rice line of inside neighbor was also planted as the border hills (red box) to avoid the margin effect on the rice growth inside the plot.

Taken together, each plot containing 280 hills, totally 1,120 hills (Page17 Line 341- line 346). Yield and its components were determined according to the method described by previous study with minor modification.

At maturity, 6-8 hills of plants from each line were selected randomly in the remaining region (see Fig. 6a and Unpublished Fig. 4) for analysis and calculation of the yield and its components.

24.-- my biggest issue with the manuscript is the controls for the phenotypic and molecular analyses. All comparisons were done relative to a wild type nipponbare. Since the rice lines were generated through agrobacterium-mediated transformation of calli a tissue culture step is involved, which typically can give rise to somaclonal variations. To avoid this, it would have been nicer to work with segregating wild types. In this way, the authors would also have been able to avoid effects due to the age of the seedstock, the different conditions for seed upscale (Temperature etc largely affect seed quality). Now, it can not be concluded if a difference in seed quality and age is not responsible for the observed phenotypes.

Response: Thank you for your important suggestion. To check effect of somaclonal variation during transformation on phenotypes in *OSA1*-overexpressing lines, we have transformed CaMV 35S empty vector into rice and investigated biomass and gas-exchange properties in this transgenic rice. As shown in Unpublished Fig. 5, there is no significant difference between WT and the empty vector transformed rice line in biomass, stomatal conductance and photosynthesis rate.

In addition, regarding the seedstock condition, we used the rice seeds of WT, *OSA1*-oxs, and *osa1* mutants that were harvested at the same time and kept at the same conditions, and their quality and germination rates were similar in each experiment. From these results, we concluded that the difference of the phenotypes was not due to somatic variation and the difference in seed quality and age. To support this conclusion, we have investigated detailed phenotypes in *osa1*-knockout mutants and obtained consistent results. Therefore, we'd like to leave as it is.

Unpublished Fig. 5 Plant growth and gas-exchange properties in CaMV 35S empty vector transformed rice. (a) Dry weights of root and shoot in WT and CaMV 35S empty vector transformed plants. Stomatal conductance (b) and CO₂ assimilation rate (c) in response to light in WT and CaMV 35S empty vector transformed plants. Error bars represent the SD (n = 3) (a to c). Differences were assessed using the Student's *t*-test (**P* < 0.05; ** *P* < 0.01).

25. I'm not a stomata expert, but I'm puzzled by Fig.4: Is the right panel of the OE line (legend suggests that) or an example of an open stoma (figure suggests that)? More evidence should be provided to conclude that there is no difference in morphology. The text now states that because there is no difference in density and size that there is no effect on morphology: I do not fully agree: morphology can be altered while density stays the same.

Response: Thank you very much pointing this out. We are very sorry that we miswrote the legend in Fig. 4a. Both pictures are from wild type stomata. We revised corresponding Figure legend and text.

Regarding morphology, thank you very much for your important suggestion. We provided the pictures of stomata from WT, *OSA1*-oxs, and *osa1* mutant plants (Supplementary Fig. 3c). As you can see, there is no difference in stomatal shape between WT and *OSA1*-oxs. These results strongly suggested that overexpression of PM H⁺-ATPase in rice had no effect on stomatal morphology and development. Therefore, we changed 'indicating' into 'suggesting', but we'd like to leave the following sentence as it is.

26. the data are very descriptive: we OE a gene and this is what we see, without providing mechanistic insights. The authors attempt to do so by providing new data in the

discussion, but this is limited (and should be in the results section and not in the discussion).

Response: we have provided mechanistic insights in the reversion according to your suggestions and explained in followings:

To study on PM H⁺-ATPase in relation to the NH₄⁺ uptake and assimilation in paddy rice with stomata opening in guard cells, we have analyzed various physiological and molecular mechanisms and also the field trials with many agronomic parameters. Furthermore, we performed RNAseq analysis and identified the genes involved in rice plants by modification of *OSA1*. We have provided these data in the Results Section.

Stomatal opening, caused by overexpression of PM H⁺-ATPase gene, leads to higher transpiration, which in turn enhances water uptake with various nutrients. However, ammonium is toxic when it is excessively taken up by plants, including rice. We found that under ammonium nutrition, overexpression of *AHA2* in guard cells caused severer ammonium toxicity in growth of *Arabidopsis*, compared to WT (Unpublished Fig. 6). This should be induced by accelerated uptake of ammonium due to the higher transpiration in *AHA2* overexpression *Arabidopsis*.

Besides the evidences that ammonium uptake increased in rice plants by enhanced activity of PM H⁺-ATPase in WT plants by fusicoccin or by overexpression of rice *OSA1* genes, actually we also used PM H⁺-ATPase inhibitor vanadate and mutants of *OSA1* gene to convince this outcome.

Furthermore, this time we checked the ammonium uptake under various ammonium concentrations from 0.5–8 mM by over expression lines and mutants. The results showed that overexpression of *OSA1* had higher absorption rate of ammonium than WT, while mutants had lower absorption rate than WT (e.g. Page 6 Line 109 - Line 116).

Only enhancing the uptake of ammonium by gene modification, such as overexpression of rice ammonium transporter genes (*OsAMTs*), was convinced to induce ammonium toxicity in comparison with the WT under normal ammonium concentration.

In addition, only enhancing the ammonium assimilation by gene modification, such as overexpression of GS, also caused depressed growth of rice in some studies (Cai, H. et al. 2009 Plant Cell Rep, Thomsen, H. et al. 2014 Trends Plant Sci.).

It can be speculated that absorbed ammonium in root cells can be quickly transferred to glutamine. But this process will produce excessive H⁺, which is toxic to plant cells. In this way, enhanced PM H⁺-ATPase by overexpression of *OSA* gene can exposure excessive H⁺ outside the root cells, which guarantee the ammonium assimilation process. This may be more crucial for global metabolisms in rice cells (Page 12 Line 245 – Page 13 Line 271).

Taken together, overexpression of PM H⁺-ATPase has two contributions: enhancing the uptake of ammonium and maintaining the subsequent assimilation of ammonium by pumping excessive H⁺ outside the root cells.

In addition, enhanced C fixation due to the opening of stomatal by overexpression of PM H⁺ ATPase gene will provide more C skeletons for the assimilation of NH₄⁺, and *vice versa*, enhanced NH₄⁺ uptake and assimilation can provide more amino acids for the synthesis of protein in leaves for photosynthesis (Page 13 Line 278 – Page 14 Line 292).

We have added the descriptions in the Results and Discussion as follows.

e.g Page 6 Line 109 - Line 116; Page 12 Line 245 – Page 13 Line 271 and Page 13 Line 278 – Page 14 Line 292. Please also check Supplementary Fig. 10.

Cai, H. et al. 2009. Overexpressed glutamine synthetase gene modifies nitrogen metabolism and abiotic stress responses in rice. *Plant Cell Rep.* 28, 527–537.

Thomsen, H. et al. 2014. Cytosolic glutamine synthetase: a target for improvement of crop nitrogen use efficiency?. *Trends Plant Sci.* 19(10):656-663.

1/2 hoagland solution 10 days

→ 2 mM NH₄⁺ 5 days

Unpublished Fig. 6 WT and two *GC1::AHA2* overexpression lines of *Arabidopsis* were cultivated in 2 mM NH₄⁺ solution for 5 days after pre-cultivation in 1/2 Hoagland solution. The pH of solution was 5.7.

Supplementary Fig. 10

Reviewer #3 (Remarks to the Author):

The agricultural green revolution of the 1960's enhanced cereal crop yields, fed a growing world population, and was in part due to increased cultivation of semi-dwarf GRVs. However, GRVs require a high N fertilizer supply to achieve maximum yield potential. Because environmentally degrading nitrogen fertilizer use underlies current cereal yields, it is thus imperative to develop new strategies for increasing nitrogen use efficiency (NUE). In this manuscript, the authors reported that the up-regulation of the OSA1 gene could enhance carbon fixation, nitrogen assimilation and biomass production, and consequently increased grain yield in rice. The topic is interest for researchers follow trends and important development in understanding the molecular mechanisms underlying maintaining the balance of carbon (C) and nitrogen (N) in response to environmental changes. However, a number of aspects of the manuscript are not clear and several experiments need to be improved.

27., the previous studies have shown that overexpression of the genes encoding plasma membrane (PM) H⁺-ATPase promoted stomatal opening, photosynthetic C-fixation, and plant growth. In the current version of the manuscript, the authors generated the *osa1* mutant and the transgenic rice plants overexpressing of OSA1, and confirmed that OSA1 acts as a positive regulator of C-fixation, N assimilation and plant growth in rice. However, the authors only present the observations, but did not provide the molecular mechanisms that explain how OSA1 coordinately regulates C fixation in shoots, N assimilation in roots and biomass production.

Response: Thank you for the suggestion. OSA1 encodes a PM H⁺-ATPase, which is an important ion pump protein but located downstream in the signaling pathway. In order to provide potential molecular mechanisms for the understanding of the role of OSA1 in nitrogen use efficiency and grain yield in rice, we analyzed the global gene expression profiles in leaves and roots of 4-weeks-old WT, OSA1-ox and *osa1* plants by using RNA-seq analysis. We compared the leaf and root transcriptomes between WT, OSA1-oxs, and *osa1* mutant, and found the genes associated with photosynthesis, ammonium assimilation, nitrogen utilization, amino acid metabolism, and glutamate biosynthesis were significantly enriched in the up-regulated genes in OSA1-oxs, but in down-regulated genes in *osa1* mutants.

As compared with WT, OSA1-oxs could have a higher ability to pump out excessive H⁺ during the ammonium uptake and assimilation. In addition, OSA1-oxs had higher stomatal conductance and photosynthetic rate, which could supply more carbon skeleton for ammonium assimilation in roots. The upregulated expression of ammonium transporters and ammonium assimilation genes in OSA1-oxs suggests that enhanced ammonium uptake and assimilation could have a positive feedback on the related genes expression.

We have added the molecular mechanisms in the Results and Discussion.

e.g. Page 8 Line 176 – Page 9 Line 201 and Page 12 Line 248 – Page 14 Line 292. Please also check Fig. 5, Supplementary Figs. 9 and 10.

28., to investigate if OSA1-enhanced transpiration in leaves accelerates NH₄⁺ uptake in roots and grain yields, the authors could generate the OSA1-overexpressing plants under the control of guard cell-specific promoters.

Response: Thank you for your suggestion. It is unknown the guard cell-specific promoter in rice. Therefore, we have transformed pAtGC1::GFP-GUS into rice plants. However, *Arabidopsis* guard cell-specific promoter (GC1 promoter) did not show guard-cell specific

expression in rice (Unpublished Fig. 7). Therefore, it is difficult to show the contribution of transpiration in rice leaves for NH_4^+ uptake.

Instead, we grew wild-type (WT) and *GC1::AHA2* (specific overexpression of PM H^+ -ATPase in guard cells, Wang et al. 2014 PNAS) Arabidopsis plants under hydroponic condition with 2 mM NH_4^+ . Although the higher transpiration of *GC1::AHA2* plants may help to uptake more NH_4^+ , the growth of *GC1::AHA2* plants were comparable as that of WT plants (Unpublished Fig. 8). Therefore, we suspect that only enhance the NH_4^+ uptake by transpiration is not enough for improving plant growth. The significant increment of gain yield in *OSA1-oxs* rice should be attributed to improve both NH_4^+ uptake and assimilation processes. (please see description in the Supplementary Fig.10). Because enhanced PM H^+ -ATPase activity can pump out excessive H^+ , generated during NH_4^+ assimilation (Weng, L. et al. 2020 Plant Physiol. Biochem, Zhu, Y. et al. 2009 Plant Cell Environ, Yan, F. et al.1998. Plant Physiol.).

Weng, L. et al.2020. Potassium alleviates ammonium toxicity in rice by reducing its uptake through activation of plasma membrane H^+ -ATPase to enhance proton extrusion. *Plant Physiol. Biochem.*151,429–437.

Zhu, Y. et al.2009. Adaptation of plasma membrane H^+ -ATPase of rice roots to low pH as related to ammonium nutrition. *Plant Cell Environ.* 32,1428–1440.

Yan F. et al.1998. Adaptation of active proton pumping and plasmalemma H^+ -ATPase activity of corn roots to low root medium pH. *Plant Physiol.* 117, 311-319.

Unpublished Fig. 7. Arabidopsis GC1 promoter did not show guard-cell specific expression in rice leaves. Rice plants was transformed by pAtGC1::GFP-GUS (Left). Fluorescent image for GFP (Right). Arrows indicate the position of stomata.

1/2 hoagland solution 10 days

→ 2 mM NH_4^+ 5 days

Unpublished Fig. 8 (A) WT and two *GC1::AHA2* overexpression lines of *Arabidopsis* were cultivated in 2 mM NH_4^+ solution for 5 days after pre-cultivation in 1/2 Hoagland solution. The pH of solution was 5.7.

29., in Figs. 4b and 5a, the authors showed that OSA1-mediated stomatal opening was completely suppressed by ABA, indicated a link between OSA1 function and ABA signalling. In canonical ABA signalling, PP2Cs interact with and inactivate OST1, ABA binds RCAR/PYR/PYL receptors to capture PP2Cs, releasing the inhibition of OST1, consequently promoting activities of the slow-type anion channel SLAC1 and other targets in guard cells. The authors should analyze the genetic relationship between OSA1 and the key components of ABA signaling, such as OST1 and PP2C.

Response: Thank you for your important suggestion. We have shown that ABA suppressed light-induced stomatal opening and phosphorylation of PM H^+ -ATPase, which is required for activation, in guard cells from *Arabidopsis thaliana* (Hayashi et al. 2011 Plant Cell Physiol.). In addition, *abi1-1*, *abi2-1*, and *ost1-2* mutants didn't show ABA-dependent inhibition of stomatal opening and phosphorylation of PM H^+ -ATPase in guard cells, indicating that OST1 and PP2C (ABI1 and ABI2) are involved in this response. Unfortunately, there is no ABA-insensitive mutants regarding OST1 and PP2C in rice (Kim et al. 2015 Front. Plant Sci.). Basically, the early signaling components for ABA signaling pathway, such as ABA receptors (RCAR/PYR/PYL), OST1, and PP2C (ABI1 and ABI2), are almost identical between *Arabidopsis* and rice (Kim et al. 2012 J Exp Bot.). Therefore, we believe that rice also show similar results.

Hayashi et al. 2011. Immunohistochemical detection of blue light-induced phosphorylation of the plasma membrane H^+ -ATPase in stomatal guard cells. *Plant Cell Physiol.* 52, 1238-1248 .

Kim et al. Functional characterization and reconstitution of ABA signaling components using transient gene expression in rice protoplasts. *Front. Plant Sci.* 6:614.

Kim et al. A rice orthologue of the ABA receptor, OsPYL/RCAR5, is a positive regulator of the ABA signal transduction pathway in seed germination and early seedling growth. *J Exp Bot.* 63(2):1013-1024.

30, in Fig. 1, the authors should add the un-treated control samples (mock) under light conditions.

Response: Thank you for your suggestion. We have repeated this experiment with the Mock control under light. In addition, Fig.1 is now reorganized in the revision, in order to make the relationship between NH_4^+ absorption rate and PM H^+ -ATPase or transpiration rate more clearly than the previous version. In this way, the redundant data were omitted in this revision, since the activation/phosphorylation of PM H^+ -ATPase by fusicoccin has

been convinced over 20 years before (Marrè 1979 *Ann. Rev. Plant Physiol.*, Olsson et al. 1998 *Plant Physiol.*)

Marrè, E. 1979. Fusicoccin: a tool in plant physiology. Ann. Rev. Plant Physiol. 30: 273-288

Olsson, A. et al. 1998. A phosphothreonine residue at the C-terminal end of the plasma membrane H⁺-ATPase is protected by fusicoccin-induced 14–3–3 binding. Plant Physiology. 118.2: 551-555.

31, in Fig. 6, the authors should present in more details about agronomic traits.

Response: In addition to Fig. 6, we have already shown the other agronomic traits, such as Plant height (cm), Panicles number per hill, Panicles length (cm), 1000 grain weight (g), and Filled grains rate (%), in Supplementary Tables 3-6.

32, the authors performed qRT-PCR experiments and showed the up-regulation of those genes involved in N-metabolism, it will be better that the authors carried out the RNA-seq experiments using WT, *osa1* and OSA1-overexpressing plants.

Response: Thank you for your suggestion. We have analyzed the global gene expression profiles in leaves and roots of 4-week-old WT, OSA1-ox and *osa1* plants by using RNA-seq analysis. GO term of ammonium assimilation was found to be significantly enriched in the up-regulated DEGs in OSA1-oxs and down-regulated DEGs in *osa1* mutant. Six ammonium transporter genes, and some genes involved in ammonia assimilation, such as GOGAT, GS1;2 and GS2 are strongly affected by the modification of OSA1. Please check the Result section (Page 8 Line 164 - Page 9 line 201).

Reviewer #4 (Remarks to the Author):

The manuscript “Simultaneously enhancing N and C uptake drastically increases in rice yield” describes very interesting and important discoveries in Agriculture and human society.

Authors Wrote that only one gene, H⁺-ATPase overexpression can improve crop productivity due to N and C uptake improvement.

33.

It is convincing from your manuscript that Membrane hyperpolarization due to H ATPase can improve N and C uptake and as a result improve biomass.

However, I think that increase of seed production (increase of # of panicle per hill and spikelets per panicle) is not directly related with biomass increase and explain it.

Even though you can not explain the mechanism. You should describe the changes of flowering and panicle development genes expression through transcriptome analysis.

Response: Thank you for your important suggestion.

To explain the seed production/grain yield (GY) increases in cereal crops, two variables need to be analyzed by breeders: plant biomass (BM, the total above-ground biomass) and harvest index (HI), as GY can be expressed as the product of biomass and harvest index ($GY = BM \times HI$), where biomass is taken as a measure of plant final net biomass production and harvest index provides a criterion for the assessment of successful partitioning of biomass into harvestable product (Donald & Hamblin, 1976 Adv. Agron.; Hay, 1995 Annu. Appl. Biol.).

Grain yield (GY) is attributed to improvement in biomass and/or harvest index depending on the particular species (Khush, 2001 Nat. Rev. Genet.). In the case of rice, the improvements in both harvest index and biomass have all contributed to the increase of yield potential. Since 1960s, the incorporation of a recessive gene, *SD-1*, for short stature, resulted in the development and release of high-yielding rice varieties with an increase in harvest index (Evans & Fisher, 1999 Crop Sci.). In contrast, the higher yield potential of other rice cultivars was attributed to greater biomass production rather than harvest index (Khush, 2001 Nat. Rev. Genet.; Peng et al., 1999 Crop Sci.).

In our study, overexpression of *OSA1* can promote carbon-nitrogen absorption and assimilation, which in turn significantly increased biomass of *OSA1*-oxs under hydroponic conditions. Moreover, in the field trial, overexpression of *OSA1* can significantly increase GY and the total biomass production under a range of nitrogen fertilization amounts (0~300kgN/ha). We checked the harvest index (HI) in the field trials and found that HI is similar in *OSA1*-oxs and WT rice under each treatment of N fertilization amounts. In addition, under non fertilization condition or over fertilization condition, the HI in *OSA1*-oxs is little bit lower than that in WT (Unpublished Fig. 9). This result further confirmed that the grain (or seed) production in *OSA1*-oxs is attributed to greater biomass production rather than harvest index in our study.

We have described the mechanism with additional analysis and information in details in the revision, which is partially in accordance to our previous study in *Arabidopsis* (Wang et al., 2014 PNAS). Overexpression of guard cell specific *AHA2* increased biomass and grain yield of *Arabidopsis*. Compared to the green revolution cultivars that having higher HI, but low nitrogen use efficiency, our *OSA1*-oxs drastically improves the nitrogen use efficiency and reduces fertilization of N amount in the agricultural practice.

e.g. Page12 Line 236 - Page14 Line 303; Please also check the Supplementary Fig. 10.

Donald, C.M. & Hamblin, J. *The biological yield and harvest index of cereals as agronomic and plant breeding criteria*, *Adv. Agron.* 28: 361–405 (1976).

Hay, R.K.M. *Harvest index: a review of its use in plant breeding and crop physiology*, *Annu. Appl. Biol.* 126: 197–216 (1995).

Khush, G.S. *Green revolution: the way forward*, *Nat. Rev. Genet.* 2:815–822 (2001).

Evans, L.T. & Fisher, R.A. *Yield potential: its definition, measurement, and significance*, *Crop Sci.* 39 :1544–1551 (1999).

Peng, S. et al. *Yield potential trends of tropical rice since the release of IR8 and the challenge of increasing rice yield potential*, *Crop Sci.* 39:1552–1559(1999).

Wang, Y. et al. 2014. *Overexpression of plasma membrane H⁺-ATPase in guard cells promotes light-induced stomatal opening and enhances plant growth*. *Proc. Natl. Acad. Sci. USA* 111, 533–538.

Unpublished Fig. 9 Harvest index of WT and OSA1-oxs plants in the field. Plants were grown in the summer of 2017 at Nanjing-N under N-N (0 kg N/ha), L-N (100 kg N/ha), M-N (200 kg N/ha), or H-N (300 kg N/ha) fertilization. Small circles in Figure represent the data points of individual experiments and three replicates were performed. Error bars represent the SE (n = 3). Differences were assessed using the two-tailed Student's t-test (* $P < 0.05$; ** $P < 0.01$).

According to your suggestion, we have analyzed the global gene expression profiles in leaves and roots of 4-week-old WT, OSA1-ox and *osa1* plants by using RNA-seq analysis. We found that two panicle development related genes, the G1-like (G1L/ALOG) transcriptional activator TAWAWA1 (TAW1) and its target gene, the SVP-like MADS box gene MADS55, were up-regulated around 4-fold in the leaves of OSA1-ox plants. Overexpression of TAW1 can increase the number of secondary branches and promote the formation of tertiary branches, resulting in higher grain yield (Yoshida et al., 2013 PNAS). These data partially explain the increment of panicles. However, we have not detected the difference in genes related to the flowering, such as Ehd1/Hd1/ Hd3a/RFT1/ rack1 /col19/MADS14 / MADS15/ MADS 50 (Shrestha et al., 2014 Ann. Bot.).

Taken together, the results in the present study, as similar as our previous results using *Arabidopsis* (Wang et al. 2014 PNAS), at least indicated that overexpression H⁺-ATPase

didn't destroyed the mass flow from biomass to seed production.

The mechanism of stage change (from vegetation stage to reproductive stage) is a very interesting question, which is one of our future works. We are continuously working on it in details and hope to report it in the next publication. Therefore, we have not described the results of these in details in the manuscript this time. We greatly appreciate your understanding.

Yoshida, A. et al. 2013. TAWAWA1, a regulator of rice inflorescence architecture, functions through the suppression of meristem phase transition. Proc. Natl Acad. Sci. USA, 110, 767–772

Wang, Y. et al. 2014. Overexpression of plasma membrane H⁺-ATPase in guard cells promotes light-induced stomatal opening and enhances plant growth. Proc. Natl. Acad. Sci. USA 111, 533–538.

Shrestha, R. et al. 2014. Molecular control of seasonal flowering in rice, arabidopsis and temperate cereals. Ann. Bot. 114, 1445-1458

REVIEWER COMMENTS

Reviewer #1 (Remarks to the Author):

The revised manuscript is much improved. The authors have completed a number of new experiments raised in the first review. In particular, I'm satisfied that ammonium flux is enhanced in the OEX lines across both the HATS and LATS concentration ranges, suggesting a global influence on net ammonium uptake by the OEX of OSA1 (Fig 2F, Supp Fig 6). This response is supported by a stimulation in AMT gene expression and other N-metabolic genes required to transport and assimilate ammonium into amino acids. Together these responses indicate an enhancement of the N transport and metabolic pathways in the OEX lines.

The growth responses quantified using both lab-based chamber assays and field trial experiments portray a consistent positive response in plant growth, including increases in tiller number, spikelet number per panicle and overall yield. These responses increased in a N-dependent manner and could be replicated at varying latitudes and growing conditions across diverse field sites. A loss of *osa1* (lab-based experiments), reduced net N accumulation and ammonium flux in the plants and resulted in compromised growth. Whether this is a trait which has already been captured in commercial rice lines is still to be determined.

In the first review, I requested the plants be tested using nitrate N instead of ammonium N to determine if this was an ammonium specific response. The experiment (not included in the manuscript) indicated a positive growth response in the OEX OSA1 lines on nitrate N relative to the WT plants. Nitrate isn't an ideal N source for rice without a source of supplementary ammonium in the media solution (~50-100 μ M) and iron (i.e. Fe sequestrene) in the hydroponic media. Nevertheless, the positive response to nitrate, regardless of its poor N status, indicates that the OEX of OSA1 may have multiple outcomes that are not necessarily dependent solely on elevated ammonium uptake and assimilation or the subsequent cytosolic pH modification outlined in the model in Supp Fig 10. The authors have subsequently published a paper linking increased K uptake with stimulated ATPase activity and a reduction to ammonium related toxicities, possibly through the release of H⁺ out of the cytosol into the root apoplast (Wen et al 2020, Plant Phys Biochem). These recent results further confirm that ammonium transport/assimilation must not be looked at in isolation to that of other ions that would be dependent on the PM electrical potential influenced by ATPase activity.

The generic role of OSA1 activity on PM electrical potentials will have a significant impact on other electrogenic dependent transport pathways reliant on pH and or concentration gradients across the PM. Upon request, the authors did confirm that both K and P uptake fluctuates in the transgenic lines, increasing with OSA1 OEX and decreasing in the silenced lines (Supp Fig 8). This is highly relevant data indicating that other ions are important and could be underpinning the ammonium response detailed and focussed in this manuscript. The impact on K and P should also be explored in a similar manner to that of ammonium to complete this study.

The authors propose a valid model (Supp Figure 10) indicating increased OSA1 activity is important to reduce the accumulation of H⁺ in the cytoplasm derived from ammonium assimilation. Protons are exported into the apoplast of root cell walls to reduce potential toxicities. To support this model two experiments would need to be completed:

- 1) Measurements of an expected pH change (acidification relative to WT) outside the root possibly using the non-invasive MIFE method.
- 2) Intercellular localisation of OSA1 on the PM of root cortical and epidermal cells – surprisingly this hasn't been shown in the manuscript. The increase in protein content and ATPase activity (Fig 2) is fine but doesn't show the detail (membrane localisation) required to support the model. I would suggest both a native promoter and the 35Sp be used to drive GFP tagged OSA1 activities.

In summary, this is an exciting collection of experiments indicating a positive response to a PM localised ATPase and its potential impact on N-dependent growth in rice. The link with CO₂ assimilation and altered stomata function, also by OSA OEX, would suggest a C&N balance that is carefully managed by a universally expressed ATPase activity across the plant. However, the data and probably the quantity of it in this manuscript has started to unravel the central story with new actors (K, P) and potentially others contributing to the phenotypes on display. As a result, the current title and the general direction of the paper is too aligned to N & C and is missing potentially other players contributing to the results. I suspect other nutrients (especially P and K) and their impact on growth are having a role in the ammonium response. The fact that P and K increase in the OEX OSA1 lines requires their roles in the phenotypes to be evaluated to rule in or out their relationship to plant growth and yield. The paper is not limited to an increase in NUE through OSA1 overexpression.

I believe this could be achieved with a metabolomic analysis of the rice lines used in this study to classify how many other nutrients are enhanced by OSA1 OEX or in the *osa* KO's. If consistent with ammonium, then the paper should be re-written as a general nutrient response to OSA1. This doesn't take away the impact of the paper or suitability to this journal, but it does capture a more comprehensive evaluation of the data and the potential biotechnological impact that altered OSA1 activities may have on rice growth.

Further comments

Line 1: Title should include P and K

Line 36 – delete 'the'

Line 42 – delete both 'the'

Line 42 – amounts

Line 44 – 'replace and pollutes' with 'which pollutes'

Line 45 – replace plant with plants

Line 53 – add 'the' before PM

Line 55 - add 'the' before PM

Line 56 - add 'the' before PM

Line 58 – why is the shape of the stomata important to refer to it?

Line 66 – change 'molecular' to 'molecule of'

Line 67 – change 'molecular' to 'molecules of'

Line 188-190 – The fact multiple transport systems are perturbed in the altered expression of OSA1, is a red-flag that multiple systems are at play and contributing to the overall growth and yield response observed in the OEX lines.

Line 294 – delete 'the', add s to fertilizer, delete 'is' with 'are in'

Line 300 – replace demands with demand, replace 'same' with 'similar', add 's' to yield

Line 389-392 – Western blot data should be presented.

Reviewer #2 (Remarks to the Author):

The manuscript greatly improved due to the extra measurements and experiments that were performed. However, the newly added red text needs English editing. Unfortunately, some issues remain.

- the ratios are now sometimes converted into percentages, but that does not differ the tricky statistics: for example for the percentage of open stomata: the statistics are done on three

observations, so three times (it is not even clear whether this is done on independent leaves of one plant or independent plants but same leaf number?). In addition it is not clear how many cells the authors counted to determine the percentage? I'm also not convinced that a ttest on the actual percentages is suited, but I rather think these data need to be transformed (now they are limited between 0 and 100, while ttest deals with continuous data, with no limits, like a weight which can be basically between 0 and infinity).

- the control with an empty vector does not completely resolve the somaclonal variations, because that would assume that every time a different plant goes through tissue culture the same variations would occur. In addition, the empty vector control is only assessed in the growth chamber and not in the field. This would mean that differences (or absence thereof) in the growth chamber automatically translates to the same phenotypes (or absence thereof) in the field, which we all know is not the case. So the control would only suffice for the growth chamber measurements. Either way, I also understand the authors can not include the proper controls anymore in multiyear/multilocation field trials, but it is a worry about the solidity of the data. If the data do get published, it is pivotal that all information in unpublished figure 5 (and even the information on seed age etc) is added to the manuscript, to show that at least the minimum of controls were used.

- The transcriptome data start to give some mechanistic insights, but the analysis is not entirely clear. Why do the authors only show the upregulated genes in the roots and leaves of the OE and the downregulated genes in the mutant tissues? Could the opposite comparison not be equally interesting? There is very few overlap between tissues and oppositely regulated genes between mutant and OE, but there are some: what are these genes? Maybe the fact that there is low overlap between the roots and the leaves is logic given the distinct function of both organs, but there is no information on the differences. How were the GO enrichments determined and what was the significance level and how large are these categories? Some of these categories are so small that if only one gene is retrieved is represents a significant proportion of that category, but biologically it does not mean much. Are the genes in fig 5B the only ones resulting in GO enrichments, because these are few relative to the amounts of DEGs listed in the text. What is the explanation that so many different transporter types are transcriptionally upregulated by OE of OSA1? How would this work? In other words, how does the H⁺ gradient across the PM result in transcriptional activation of so many transporters? GRF4 is shown in Figure 5 but mentioned for the first time in the discussion: there should be at least one sentence referring to GRF4 in the results. Why was GRF4 chosen? Because of the recent publication or because it stood out from the data?

Reviewer #3 (Remarks to the Author):

In this revised version of the manuscript, the authors have added additional experiments and answered the questions which I addressed. Now, it is acceptable for publication.

Reviewer #4 (Remarks to the Author):

Authors solved the my comments well during revision.

Reply to Reviewers' comments:

Thank you very much for your valuable comments and suggestions. We have revised the manuscript according to the reviewer's comments and have answered all the questions. All changes to the text are written in red letters.

REVIEWER COMMENTS

Reviewer #1 (Remarks to the Author):

The revised manuscript is much improved. The authors have completed a number of new experiments raised in the first review. In particular, I'm satisfied that ammonium flux is enhanced in the OEX lines across both the HATS and LATS concentration ranges, suggesting a global influence on net ammonium uptake by the OEX of OSA1 (Fig 2F, Supp Fig 6). This response is supported by a stimulation in AMT gene expression and other N-metabolic genes required to transport and assimilate ammonium into amino acids. Together these responses indicate an enhancement of the N transport and metabolic pathways in the OEX lines.

The growth responses quantified using both lab-based chamber assays and field trial experiments portray a consistent positive response in plant growth, including increases in tiller number, spikelet number per panicle and overall yield. These responses increased in a N-dependent manner and could be replicated at varying latitudes and growing conditions across diverse field sites. A loss of *osa1* (lab-based experiments), reduced net N accumulation and ammonium flux in the plants and resulted in compromised growth. Whether this is a trait which has already been captured in commercial rice lines is still to be determined.

In the first review, I requested the plants be tested using nitrate N instead of ammonium N to determine if this was an ammonium specific response. The experiment (not included in the manuscript) indicated a positive growth response in the OEX OSA1 lines on nitrate N relative to the WT plants. Nitrate isn't an ideal N source for rice without a source of supplementary ammonium in the media solution (~50-100 μM) and iron (i.e. Fe sequestrene) in the hydroponic media. Nevertheless, the positive response to nitrate, regardless of its poor N status, indicates that the OEX of OSA1 may have multiple outcomes that are not necessarily dependent solely on elevated ammonium uptake and assimilation or the subsequent cytosolic pH modification outlined in the model in Supp Fig 10. The authors have subsequently published a paper linking increased K uptake with stimulated ATPase activity and a reduction to ammonium related toxicities, possibly through the release of H^+ out of the cytosol into the root apoplast (Wen et al 2020, Plant Phys Biochem). These recent results further confirm that ammonium transport/assimilation must not be looked at in isolation to that of other ions that would be dependent on the PM electrical potential influenced by ATPase activity.

The generic role of OSA1 activity on PM electrical potentials will have a significant impact on other electrogenic dependent transport pathways reliant on pH and or concentration gradients across the PM. Upon request, the authors did confirm that both K and P uptake fluctuates in the transgenic lines, increasing with OSA1 OEX and decreasing in the silenced lines (Supp Fig 8). This is highly relevant data indicating that other ions are important and could be underpinning the ammonium response detailed and focused in this manuscript. The impact on K and P should also be explored in a similar manner to that of ammonium to complete this study.

The authors propose a valid model (Supp Figure 10) indicating increased OSA1 activity is important to reduce the accumulation of H^+ in the cytoplasm derived from ammonium assimilation. Protons are exported into the apoplast of root cell walls to reduce potential

toxicities. To support this model two experiments would need to be completed:

1) Measurements of an expected pH change (acidification relative to WT) outside the root possibly using the non-invasive MIFE method.

Response: Thank you very much for your important suggestion. To determine pH change outside the root, first, we monitored the rhizosphere acidification using agar plates containing bromocresol purple as a pH indicator (Yan et al., 2002; Zhu et al., 2005). Please see **Supplementary Fig. 2a-e**. It is obvious that *OSA1*-overexpressing lines (*OSA1*-oxs) released significantly higher amount of H⁺ from roots, compared to the wild type (WT) roots. But the seminal root surface area, length and number in *OSA1*-oxs are almost same as in WT at this age.

In addition, we tried to determine the H⁺ efflux rates from roots by non-invasive MIFE method. As shown in **Supplementary Fig. 2f**, *OSA1*-oxs showed higher H⁺ efflux rates than WT from 1.5 min after the start of measurement. Taken together, our results are totally in agreement with our hypothesis, same as your prediction. We described these results in the text as follows.

Page 5, line 102- Page 6, 104; We confirmed higher H⁺ extrusion from roots in *OSA1*-oxs (Supplementary Fig. 2) and proper localisation of overexpressed PM H⁺-ATPase in roots (Supplementary Fig. 3).

Yan, F, et al., 2002. Adaptation of H⁺-pumping and plasma membrane H⁺-ATPase activity in proteoid roots of white lupin under phosphate deficiency. Plant Physiol. 129, 50–63.

Zhu, Y, et al., 2005. A link between citrate and proton release by proteoid roots of white lupin (Lupinus albus L.) grown under phosphorus-deficient conditions? Plant Cell Physiol. 46: 892-901

Supplementary Fig. 2. Rhizosphere acidification and H⁺ efflux properties of WT and OSA1-oxs rice. (a) Monitoring of rhizosphere acidification around WT and OSA1-oxs rice roots. Plants were grown in nutrient solution for 7 days. After washing with deionized water, roots were carefully spread onto solid medium containing 0.02% (w/v) bromocresol purple and 0.7% (w/v) agar adjusted to pH 6.5. After incubation for 12 hr in the dark, the plates were photographed. (b) Relative area of rhizosphere acidification (yellow area) on the surface of agar plates. (c-e) Surface area (c), length(d) and number (e) of seminal roots in WT and OSA1-oxs. Small circles represent the data points of individual experiments were performed. Values are mean \pm SEs (n = 5). **f** Quantification of H⁺ efflux rates from WT and OSA1-oxs roots. Intact roots (3-5 cm in length from the root tips) were equilibrated in the measuring solution for 10 min. Values are mean \pm SEs (n = 6). Differences were evaluated using the two-tailed Student's *t*-test (**P* < 0.05; ***P* < 0.01, n.s., not significant).

Intercellular localization of OSA1 on the PM of root cortical and epidermal cells – surprisingly this hasn't been shown in the manuscript. The increase in protein content and ATPase activity (Fig 2) is fine but doesn't show the detail (membrane localization) required to support the model. I would suggest both a native promoter and the 35S p be used to drive GFP tagged OSA1 activities.

Response: Thank you for your suggestion. It has been reported that the N-terminus tagged-PM H⁺-ATPases don't function normally (Lanfermeijer et al. 1998, Ekberg et al. 2010). Therefore, we performed the immunohistochemical staining using polyclonal antibodies of PM H⁺-ATPase and checked the localization of PM H⁺-ATPase in rice roots. We found that the fluorescence signals by anti-H⁺-ATPase antibody were observed on the plasma membrane of wild type root cortex, epidermis (EP), exodermis (EX) and endodermis (EN) (**Supplementary Fig. 3a-e**). Furthermore, the localization of PM H⁺-ATPase in roots of OSA1-oxs is the same as those in wild type (**Supplementary Fig. 3f-t**). In addition, OSA1-oxs showed around 50% more PM H⁺-ATPase protein amount, and around 30% higher PM H⁺-ATPase activity than WT (**Fig. 2d, e**). Taken together, these results indicate that the overexpressed PM H⁺-ATPase in OSA1-oxs localizes on the plasma membrane and works properly. We described these results in the text as follows.

Page 5, line 102- Page 6, 104; We confirmed higher H⁺ extrusion from roots in OSA1-oxs (Supplementary Fig. 2) and proper localisation of overexpressed PM H⁺-ATPase in roots (Supplementary Fig. 3).

Supplementary Fig. 3. Localization of PM H⁺ ATPase in rice roots. Immunohistochemical staining of 3-weeks-old seedlings in WT and OSA1-oxs, with polyclonal antibodies recognizing rice PM H⁺-ATPase (anti-H⁺-ATPase) was performed in rice root. Red color shows the signal of anti-H⁺-ATPase (**b, j, l and q**), blue color shows autofluorescence of cell wall stained by DAPI (**a, f, k and p**), and the merged image (**c-e, h-j, m-o and h-j**). EP, epidermis; EX, exodermis; EN, endodermis; PH, phloem; XY, xylem. Bars = 150 μ m (**a-c, f-h, k-m and p-r**), 20 μ m (**d, i, n and s**) and 60 μ m (**e, j, o and t**).

Lanfermeijer FC, Venema K, Palmgren MG (1998) Purification of a histidine-tagged plant plasma membrane H⁺-ATPase expressed in yeast. *Protein Expression Purification* 12: 29–37

Ekberg K, Palmgren MG, Veierskov BBuch-Pedersen MJ. (2010) A novel mechanism of P-type ATPase autoinhibition involving both termini of the protein. *J. Biol. Chem.* 285: 7344–7350

In summary, this is an exciting collection of experiments indicating a positive response to a PM localized ATPase and its potential impact on N-dependent growth in rice. The link with CO₂ assimilation and altered stomata function, also by OSA OEX, would suggest a C&N balance that is carefully managed by a universally expressed ATPase activity across the plant. However, the data and probably the quantity of it in this manuscript has started to unravel the central story with new actors (K, P) and potentially others contributing to the phenotypes on display. As a result, the current title and the general direction of the paper is too aligned to N & C and is missing potentially other players contributing to the results. I suspect other nutrients (especially P and K) and their impact on growth are having a role in the ammonium response. The fact that P and K increase in the OEX OSA1 lines requires their roles in the phenotypes to be evaluated to rule in or out their relationship to plant growth and yield. The paper is not limited to an increase in NUE through OSA1 overexpression.

I believe this could be achieved with a metabolomic analysis of the rice lines used in this study to classify how many other nutrients are enhanced by OSA1 OEX or in the *osa* KO's. If consistent with ammonium, then the paper should be re-written as a general nutrient response to OSA1. This doesn't take away the impact of the paper or suitability to this journal, but it does capture a more comprehensive evaluation of the data and the potential biotechnological impact that altered OSA1 activities may have on rice growth.

Response: We really appreciate for the reviewer's suggestion about this point. We agree that the *OSA1-oxs* improve uptake of various nutrients. Actually, we found that the content of most nutrients in *OSA1-oxs* plants are higher than those in WT. We showed these data as **Supplementary Fig. 6**.

It is also logic that uptake of nutrients in roots may be enhanced through the higher H⁺ gradient across the root cell membrane by PM H⁺-ATPase. It is unfortunately that we can't get the radioactive isotope P and K from abroad due to the COVID19 pandemic. So, we can't conduct the test of uptake rate by roots.

Furthermore, we found that, in comparison with P and K, N had the much greater effect on the promotion of biomass (please see **Unpublished Fig.1**) (Shang et al., 2014. *Plant Soil*; Marschner, 2012. *Marschner's Mineral Nutrition of Higher Plants*; and our own data from field experiments). It is clear that N fertilizer has a constant positive effect on the grain yield, while P and K fertilizer have positive effects on grain yield only with a limited range. These results indicate that the N is the major limit factor for the grain yield of rice. Therefore, we would like to leave the context as it is, but we change the title as follows.

“Overexpression of plasma membrane H⁺-ATPase *OSA1* increases rice grain yield via simultaneous enhancement of nutrient uptake and photosynthesis”

Unpublished Fig. 1 Rice yield based on the input of N, P, K fertilization (Shang et al., 2014. *Plant Soil*; Marschner, 2012. *Marschner's Mineral Nutrition of Higher Plants*; and our own data from field experiments).

Shang Q, Ning L, Feng Xu, Yang X, Wu P, Zou J, Shen Q, Guo S. (2014) Soil fertility and its significance to crop productivity and sustainability in typical agroecosystem: a summary of long-term fertilizer experiments in China. *Plant Soil* 381:13-23

Marschner, P. *Marschner's Mineral Nutrition of Higher Plants* chapter 5 (Elsevier Ltd., 2012).

Further comments

Line 1: Title should include P and K

Response: Thank you very much. As we described above, we have corrected the title according to your suggestions in Page 1, line 1 - line 2;

“Overexpression of plasma membrane H⁺-ATPase OSA1 increases rice grain yield via simultaneous enhancement of nutrient uptake and photosynthesis”

Line 36 – delete ‘the’

Line 42 – delete both ‘the’

Line 42 – amounts

Line 44 – ‘replace and pollutes’ with ‘which pollutes’

Line 45 – replace plant with plants

Line 53 – add ‘the’ before PM

Line 55 - add ‘the’ before PM

Line 56 - add ‘the’ before PM

Response: Thank you very much. We have corrected the text according to your suggestions.

Line 58 – why is the shape of the stomata important to refer to it?

Response: We deleted the description about stomatal shape.

Line 66 – change ‘molecular’ to ‘molecule of’

Line 67 – change ‘molecular’ to ‘molecules of’

Response: Thank you very much. We have corrected the text according to your suggestions.

Line 188-190 – The fact multiple transport systems are perturbed in the altered expression of OSA1, is a red-flag that multiple systems are at play and contributing to the overall growth and yield response observed in the OEX lines.

Response: It is true, PM H⁺-ATPase is important for various nutrient uptake and translocation by providing membrane potential and proton motive force (Palmgren et al. 2001). Actually, the multiple transport systems, induced by OSA1 overexpression and downregulated by *osa1* mutation (**Fig. 5b**) could also be caused by the different transpiration rate or different growth status of rice plants, which have the different requirement for overall nutrients. Although the exact mechanism is not totally clear, we think these consequent responses are the secondary effect after the OSA1 overexpression or mutation in rice. It would be interesting to investigate the underlying molecular mechanisms, how PM H⁺-ATPase activity affects the expression levels of nutrient transporters, in the future. We had mentioned in the text as follows.

Page 10, line 211-212; These results suggest a potential role for OSA1 in modulating ion and solute transport in plants.

Palmgren, M.G. 2001. Plant plasma membrane H⁺-ATPase: Powerhouses for nutrient uptake. Annu. Rev. Plant Physiol. Plant Mol. Biol. 52: 817-854

Line 294 – delete ‘the’, add s to fertilizer, delete ‘is’ with ‘are in’

Line 300 – replace demands with demand, replace ‘same’ with ‘similar’, add ‘s’ to yield

Response: Thank you very much. We have corrected the text according to your suggestions.

Line 389-392 – Western blot data should be presented.

Response: Here, I'd like to show typical Western blot data (**Unpublished Fig. 2**). In Nature Communications, we will present all Western blot data in Source data. So, we'd like to leave as it is.

Unpublished Fig. 2 Relative PM H⁺-ATPase protein levels in WT, OSA1-ox and *osa1* mutant plants.

Reviewer #2 (Remarks to the Author):

The manuscript greatly improved due to the extra measurements and experiments that were performed. However, the newly added red text needs English editing. Unfortunately, some issues remain.

Response: The English in revised manuscript has been checked by at least two professional editors, both native speakers of English. For a certificate, please see:

<http://www.textcheck.com/certificate/06MOWX>

- the ratios are now sometimes converted into percentages, but that does not differ the tricky statistics: for example for the percentage of open stomata: the statistics are done on three observations, so three times (it is not even clear whether this is done on independent leaves of one plant or independent plants but same leaf number?). In addition it is not clear how many cells the authors counted to determine the percentage? I'm also not convinced that a test on the actual percentages is suited, but I rather think these data need to be transformed (now they are limited between 0 and 100, while test deals with continuous data, with no limits, like a weight which can be basically between 0 and infinity).

Response: Thank you for your suggestion. Since the space of figure legend is limited, we didn't show the details there. Instead, the details of stomatal observation experiment had been described in Methods section, page 22, line 489 to 493, in the text. According to your comment, we added "for the details see Methods" in the figure legend to prompt reader. We use the open stomata percent based on the total 100 stomata to calculate the percent in our research. It is no doubt that t-test can be used as statistical test of percentages.

- the control with an empty vector does not completely resolve the somaclonal variations, because that would assume that every time a different plant goes through tissue culture the same variations would occur. In addition, the empty vector control is only assessed in the growth chamber and not in the field. This would mean that differences (or absence thereof) in the growth chamber automatically translates to the same phenotypes (or absence thereof) in the field, which we all know is not the case. So the control would only suffice for the growth chamber measurements. Either way, I also understand the authors can not include the proper controls anymore in multiyear/multilocation field trials, but it is a worry about the solidity of the data. If the data do get published, it is pivotal that all information in unpublished figure 5 (and even the information on seed age etc) is added to the manuscript, to show that at least the minimum of controls were used.

Response: We are very sorry that we cannot perform field experiments with empty vector. But according to your suggestion, we showed the results from empty vector transformed control line under hydroponic conditions as a **Supplementary Fig. 4** (previously Unpublished Fig. 5), which include the growth, the relative expression of *OSA1* gene, the PM H⁺-ATPase protein levels, stomatal opening (stomatal conductance), and photosynthesis rate. In addition, we used same age seeds for these experiments. We added these results in the text as follows.

Page 6, line 108-111; We observed no significant phenotype changes related to growth, relative *OSA1* gene expression, PM H⁺-ATPase protein levels, stomatal opening (stomatal conductance) and photosynthesis rate in the empty vector-transformed rice under hydroponic conditions (Supplementary Fig. 4).

Page 16, line 351-351; The rice seeds for these experiments were of the same age.

Moreover, we have already showed the phenotypes of *osa1* mutants (*osa1-1* to *osa1-3*, TOS17 insertional mutants) (**Fig. 3a** and **Supplementary Fig. 1**). As compared to WT, *osa1* mutants showed 33–52% lower dry weight than WT in growth chamber (**Fig. 3a, b**), and the NH_4^+ and CO_2 uptake also show lower than in WT. In addition, in this revision, we provided the data of the *osa1* mutants in the field trials (**Supplementary Tables 3–5**). We described these results in the text as follows.

Page 11, line 236-238; Conversely, in *osa1* mutants, grain yield was significantly lower than that of the WT at all three locations (Supplementary Tables 3–5).

These results strongly suggest that phenotypes in *OSA1*-oxs depend on expression level and activity of PM H^+ -ATPase, but not from the somaclonal variations. We hope you accept our evidence.

- The transcriptome data start to give some mechanistic insights, but the analysis is not entirely clear. Why do the authors only show the upregulated genes in the roots and leaves of the OE and the downregulated genes in the mutant tissues? Could the opposite comparison not be equally interesting?

Response: Thank you for your important suggestion. According to your comments, we analyzed the global gene expression profiles in leaves and roots of 4-weeks-old WT, *OSA1*-ox and *osa1* plants obtained by RNA-seq analysis. We compared the leaf and root transcriptomes between WT, *OSA1*-ox, and *osa1* mutant, and found genes associated with nucleic acid binding transcription factor activity, response to chitin, response to organonitrogen compound, regulation of nitrogen compound metabolic process, regulation of nucleobase-containing compound metabolic process and RNA biosynthetic process were significantly enriched in the overlapped genes down-regulated in *OSA1*-ox leaves and up-regulated in *osa1* mutant leaves ($FDR < 0.05$) (**Supplementary Table 9**). However, no GO terms were found to be significantly enriched in the overlapped genes down regulated in *OSA1*-ox roots and upregulated in *osa1* mutant roots (**Supplementary Table 9**). We described these results in the text as follows.

Page 9, line 195-203; In addition, we compared leaf and root transcriptomes between the WT, *OSA1*-ox line, and *osa1-2* mutant, and found that genes associated with nucleic acid binding transcription factor activity, response to chitin, response to organonitrogen compound, regulation of N compound metabolic process, regulation of nucleobase-containing compound metabolic process, and RNA biosynthetic process were significantly enriched in the overlapping genes downregulated in *OSA1*-ox leaves and upregulated in *osa1* mutant leaves (false discovery rate [FDR] < 0.05) (Supplementary Table 9). However, no GO terms were found to be significantly enriched in overlapping genes downregulated in *OSA1*-ox roots and upregulated in *osa1-2* mutant roots (Supplementary Table 9).

There is very few overlap between tissues and oppositely regulated genes between mutant and OE, but there are some: what are these genes? Maybe the fact that there is low overlap between the roots and the leaves is logic given the distinct function of both organs, but there is no information on the differences.

Response: Thank you for your important suggestion. According to your comments, we also analyzed the overlapped genes between root DEGs and shoot DEGs that were oppositely regulated between *OSA1*-ox and *osa1* mutant, and found that genes

associated with transmembrane transporter activity, ion transport, substrate-specific transmembrane transporter activity, cation transmembrane transporter activity, carbohydrate transmembrane transporter activity and PM part were significantly enriched in overlapping genes that were upregulated in OSA1-ox roots and leaves, but downregulated in the *osa1-2* mutant (Supplementary Table 10). We described these results in the text as follows.

Page 9, line 190-195; Genes associated with transmembrane transporter activity, ion transport, substrate-specific transmembrane transporter activity, cation transmembrane transporter activity, carbohydrate transmembrane transporter activity, and PM part were also significantly enriched in overlapping genes that were upregulated in OSA1-ox roots and leaves, but downregulated in the *osa1-2* mutant (Supplementary Table 10).

How were the GO enrichments determined and what was the significance level and how large are these categories? Some of these categories are so small that if only one gene is retrieved it represents a significant proportion of that category, but biologically it does not mean much. Are the genes in fig 5B the only ones resulting in GO enrichments, because these are few relative to the amounts of DEGs listed in the text.

Response:

Thank you for your suggestion and pointing this out. We used GO Term Enrichment tool in the Plant Transcriptional Regulatory Map (PlantRegMap) website (<http://plantregmap.gao-lab.org/go.php>) (Tian et al. 2020). GO category (<http://geneontology.org/>) false discovery rate (FDR) ≤ 0.05 was regarded as significantly enriched. GO term enrichment analysis of the DEGs upregulated in OSA1-ox and downregulated in *osa1-2* mutant was performed to investigate the molecular mechanisms underlying OSA1-mediated biological processes (**Supplementary Fig. 9d, e** and **Supplementary Table 8**). We described these results in the text as follows.

Page 24, line 537-541; GO term enrichment was conducted using GO Term Enrichment tool in the Plant Transcriptional Regulatory Map (PlantRegMap) website⁶¹ (<http://plantregmap.gao-lab.org/go.php>). GO category (<http://geneontology.org/>) false discovery rate (FDR) ≤ 0.05 was regarded as significantly enriched.

In addition, in this revision, we provided the data of all enriched GO terms include at least 2 genes (**Supplementary Table 8**). Because of the limited space, only some representative GO terms enriched were shown in **Fig. 5b**. All the differentially expressed genes belonging to the corresponding enriched GO terms were listed in **Fig. 5b**. But we have to mention that there are overlaps between different GO terms. In order to avoid the confusion, we used gene functions instead of GO terms to classify these gene (**Fig. 5b**). We described these in the **Fig.5b** as follows.

Page 35, line 785-786; Significant genes in response to N ,C metabolism and ion transport are listed (FDR < 0.05).

⁶¹ Tian, F., Yang, D., Meng, Y, Q., Jin, J., Gao, G. 2020. PlantRegMap: charting functional regulatory maps in plants, *Nucleic Acids Research*, 48, D1104–D1113.

What is the explanation that so many different transporter types are transcriptionally upregulated by OE of OSA1? How would this work? In other words, how does the H⁺ gradient across the PM result in transcriptional activation of so many transporters?

Response: Thank you for your important suggestion and pointing this out. It is very interesting that many different ion transporter genes are upregulated in *OSA1* ox (**Fig. 5b**). Actually, we found that increased contents of N, P, K, and other nutrients as compared with control plants (**Supplementary Fig. 6**). As you know, PM H⁺-ATPase is important for various nutrient uptake and translocation (Palmgren et al. 2001). Although the exact mechanism is not clear, it would be interesting to investigate the underlying molecular mechanisms, how higher PM H⁺-ATPase activity affects the expression levels of nutrient transporters, in the future. We had mentioned in the text as follows.

Page 10, line 211-212; These results suggest a potential role for *OSA1* in modulating ion and solute transport in plants.

Palmgren, M.G. 2001. Plant plasma membrane H⁺-ATPase: Powerhouses for nutrient uptake. Annu. Rev. Plant Physiol. Plant Mol. Biol. 52: 817-854

GRF4 is shown in Figure 5 but mentioned for the first time in the discussion: there should be at least one sentence referring to GRF4 in the results. Why was GRF4 chosen? Because of the recent publication or because it stood out from the data?

Response: Thank you for your important suggestion. According to your comments, we now describe it in the results section. We described these results in the text as follows.

Page 10, line 221-222; Notably, *GRF4*, a key transcription factor in N metabolism and C fixation in rice³¹, was highly expressed in response to *OSA1* overexpression (**Fig. 5c**).

³¹*Li, S. et al. 2018. Modulating plant growth–metabolism coordination for sustainable agriculture. Nature 111, 595–560.*

REVIEWERS' COMMENTS

Reviewer #1 (Remarks to the Author):

I've reviewed the re-submitted manuscript carefully. The responses to the majority of my queries raised against the second submission have been met. This edition is significantly stronger, balanced and delivers a substantial contribution to understanding the role of H⁺-ATPASE activity on nutrient transport into and within plant cells. The inclusion of new experiments (pH plates, MIFE measurements of H⁺ release and cellular localisation of ATPase proteins in roots) provides important controls and context to the manuscript. I'm satisfied with the current version and happy to support its move towards publication.

Reviewer #2 (Remarks to the Author):

- The manuscript improved by the additional information, but the data were mainly added to the results and the authors failed to bring the findings together in the discussion. For instance, the authors now show that OSA1-overexpressing lines released significantly higher amount of H⁺ from roots, compared to the wild type (WT) roots. At that time it does not result in phenotypic alterations? So what would be the effect of higher H⁺? Is this directly or indirectly effecting growth, stomatal closure or photosynthesis? What is known in literature? Similar for the massive transcriptional upregulation of the transporters: was this already previously described? How/why would OE of an ATPase result in such specific transcriptional readout? Are there transcriptional changes that might explain the differences in stomatal conductance or photosynthesis. I understand that the exact mechanistics still need to be examined, but it would be nice to read what is known in literature or what is the working hypothesis of the authors. In the rebuttal letter, the authors state that the data support their hypothesis, so it would be nice to see this hypothetical mode of action reflected in the discussion. The data are now described but not sufficiently put in context.
- Are the *osa1* mutants in the same genetic background as the OE lines (all nipponbare?)? What are the effects of the mutations? 1 WT for all these lines? And they were all grown meticulously to have the same age, always together to avoid environmental effects? It remains really cumbersome and should definitely be avoided.
- Are the RNAseq data deposited? MIAMI compliant?

Reply to Reviewers' comments:

Thank you very much for your valuable comments and suggestions. We have revised the manuscript according to the reviewer's comments and have answered all the questions. All changes to the text are shown by track changes.

Reviewer #1 (Remarks to the Author):

I've reviewed the re-submitted manuscript carefully. The responses to the majority of my queries raised against the second submission have been met. This edition is significantly stronger, balanced and delivers a substantial contribution to understanding the role of H⁺-ATPASE activity on nutrient transport into and within plant cells. The inclusion of new experiments (pH plates, MIFE measurements of H⁺ release and cellular localisation of ATPase proteins in roots) provides important controls and context to the manuscript. I'm satisfied with the current version and happy to support its move towards publication.

Response: Thank you very much.

Reviewer #2 (Remarks to the Author):

- The manuscript improved by the additional information, but the data were mainly added to the results and the authors failed to bring the findings together in the discussion.

For instance, the authors now show that OSA1-overexpressing lines released significantly higher amount of H⁺ from roots, compared to the wild type (WT) roots. At that time it does not result in phenotypic alterations? So what would be the effect of higher H⁺? Is this directly or indirectly effecting growth, stomatal closure or photosynthesis? What is known in literature?

Response: Thank you for the comments, we now have modified the discussion according to your suggestions in more detail as possible as we can.

For instance, the OSA1-ox lines released significantly higher amount of H⁺ from roots, compared to the wild type (WT) roots. This result caused stronger acidification in OSA1-ox rice roots zone, which could provide higher proton motive force in the rhizosphere for the uptake of nutrients. This is consistent with the enhanced NH₄⁺ uptake rate in OSA1-oxs as compared with WT plants (Fig. 2f,g; Supplementary Fig. 6). Since it is a local effect of overexpression of OSA1 in roots, we don't think it will influence the photosynthesis and stomatal movement directly.

We have discussed from the lines 278-281 and 313-320 in the revision.

Similar for the massive transcriptional upregulation of the transporters: was this already

previously described? How/why would OE of an ATPase result in such specific transcriptional readout? Are there transcriptional changes that might explain the differences in stomatal conductance or photosynthesis. I understand that the exact mechanistic still need to be examined, but it would be nice to read what is known in literature or what is the working hypothesis of the authors. In the rebuttal letter, the authors state that the data support their hypothesis, so it would be nice to see this hypothetical mode of action reflected in the discussion. The data are now described but not sufficiently put in context.

Response: Thank you for your important comments, we have discussed the transcriptional upregulation of nutrient transporter genes, such as ammonium transporters, phosphate transporter, and potassium transporter in response to the *OSA1*-overexpression in roots. Please check the lines 278-281 and 313-320. We have also discussed the transcriptional changes that might explain the differences in photosynthesis from lines 263 and 302-305. Regarding stomatal conductance, we think overexpression of PM H⁺-ATPase *OSA1* has simply contributed for enhanced stomatal opening. So, we don't explain the difference stomatal conductance here. However, it will be needed to investigate transcriptional changes in guard cells from *OSA1*-oxs.

In addition, we also discussed the GRF4, a transcription factor in rice, which was reported to integrate N assimilation, C fixation, and plant growth. Since GRF4 was found to be upregulated by *OSA1* overexpression (Fig. 5b, c). It is likely that some master regulators of nutrient uptake and metabolism, such as GRF4, could be activated by *OSA1* overexpression. The enhanced C fixation and N metabolism could also have a feedback on the expression of nutrient transporter genes in order to ensure sufficient supply of nutrients for the promotion of plant growth. Further study is deserved to investigate the underlying molecular mechanisms responsible for the signal transduction initiated by *OSA1* overexpression. We have discussed from the line 322-331 in the revision.

- Are the *osa1* mutants in the same genetic background as the OE lines (all nipponbare)? What are the effects of the mutations? 1 WT for all these lines? And they were all grown meticulously to have the same age, always together to avoid environmental effects? It remains really cumbersome and should definitely be avoided.

Response: Yes, *osa1* mutants are the same genetic background (*Oryza sativa* L. ssp. *japonica* cv. Nipponbare) as the OE lines.

osa1 mutants are the insertional mutant of TOS17 into the coding region of *OSA1* gene, resulting in knockout of the target gene (null mutant) (line 101; Supplementary Figure 1a).

The rice seeds for all these experiments were from the same batch, the same age and the same grown environment. We have described the related information in Methods (Plant cultivation section).

- Are the RNAseq data deposited? MIAMI compliant?

Response: We have deposited in the DNA Data Bank of Japan (DDBJ) and provided accession number DRA011260 in the Data availability..